# Ranking Startups Using DEMATEL-ANP-Based Fuzzy PROMETHEE II

Huyen Trang Nguyen [1] and Ta-Chung Chu [2,*]

1    College of Business, Southern Taiwan University of Science and Technology, Tainan City 710301, Taiwan; da51g207@stust.edu.tw
2    Department of Industrial Management and Information, Southern Taiwan University of Science and Technology, Tainan City 710301, Taiwan
*    Correspondence: tcchu@stust.edu.tw

**Abstract:** In entrepreneurship management, the evaluation and selection of startups for acceleration programs, especially technology-based startups, are crucial. This process involves considering numerical and qualitative criteria such as sales, prior startup experience, demand validation, and product maturity. To effectively rank startups based on the varying importance of these criteria, a fuzzy multi-criteria decision-making (MCDM) approach is needed. Although MCDM methods have been successful in handling complex problems, their application in startup selection and evaluating criteria interrelationships from the accelerator perspective is underexplored. To address this gap, a hybrid DEMATEL-ANP-based fuzzy PROMETHEE II model is proposed in this study, facilitating startup ranking and examining interrelationships among factors. The resulting preference values are fuzzy numbers, necessitating a fuzzy ranking method for decision-making. An extension of ranking fuzzy numbers using a spread area-based relative maximizing and minimizing set is suggested to enhance the flexibility of existing ranking MCDM methods. Algorithms, formulas, and a comparative analysis validate the proposed method, while a numerical experiment verifies the viability of the hybrid model. The final ranking of four startup projects is $A_4 < A_1 < A_3 < A_2$ which indicates that startup project $A_2$ has the highest comprehensive potential, followed by startup project $A_3$.

**Keywords:** DEMATEL; ANP; PROMETHEE II; ranking fuzzy numbers; startups

**MSC:** 91B06

## 1. Introduction

Entrepreneurship has been recognized as a significant driver of economic growth, both directly and indirectly, and as well as a catalyst for more investments in knowledge creation and generation [1]. Notably, technology-based startups can transform the traditional economy into a digital economy through innovation [2]. The key determinants of entrepreneurial success encompass a range of factors, including entrepreneurs' networks, leadership skills, financial competency, aptitude, knowledge, and support services [3]. Stam [3] defined the entrepreneurial ecosystem as a complex network of interconnected actors and factors that collaborate to facilitate productive entrepreneurship. Among the factors, accelerators are the primary players in the entrepreneurial ecosystem and are actively engaged in fostering innovation and nurturing startups. They develop startup projects, including financing, services, networking, mentoring, and training [4]. Not only do accelerators support through networking services, mentorships, and educational endeavors, but they also play a crucial role in augmenting the financial capabilities of entrepreneurial firms. However, despite their critical role, exploring the selection process employed by accelerators in identifying and evaluating entrepreneurial firms and the underlying selection criteria remains relatively scarce [5].

The initial phase in the process of the entry-boost-exit process is to select a suitable startup. Accelerators whose financial gains are contingent upon the successful exit of the startups in which they invest must exercise discernment when evaluating prospective projects [5]. The selection process encompasses three distinct steps: soliciting startup submissions, conducting comprehensive examinations and evaluations of the projects, and, based on the input of key decision-makers (DMs), eliminating unpromising projects while investing in those that exhibit promise [6]. Lin et al. [7] used the hesitant fuzzy linguistic (HFL) multi-criteria decision-making (MCDM) method to evaluate startups from a technology business incubator perspective, taking into account DMs' psychology. The researchers developed a ratio of score value to deviation degree to compare HFL term sets and defined the HFL information envelopment efficiency, analysis, and preference model. Their numerical example showed the method's applicability, and they concluded that it is more flexible and general. Nonetheless, it should be noted that this method exclusively applies to HFL information environments with unrevealed criteria weight values. Furthermore, the authors acknowledged the limited extent of research on ranking startups within the existing literature.

The process of selecting startups for acceleration programs involves intricate consideration of both qualitative and quantitative criteria. Qualitative criteria encompass factors such as competitive advantage and demand validation, while quantitative criteria include investment costs and team size. Consequently, the ranking of startups poses an MCDM problem. MCDM, as a research field, contributes to the development and implementation of decision-support methodologies and tools [8]. Additionally, MCDM methods are valuable in resolving multiplex problems involving objectives, multiple criteria, and alternatives rated by DMs. It is important to note that the DMs' judgment through qualitative criteria is crucial to the decision-making process, despite its inherent subjectivity and vagueness. Fuzzy numbers (FNs) offer a more effective means of modeling human thought compared to their crisp counterparts.

However, the conventional MCDM method solely adheres to classical mathematical theory, and different methods must be improved or combined to adapt to actual MCDM [9]. Moreover, the amalgamation of DEMATEL-ANP-based fuzzy PROMETHEE II has not been previously applied. This study aims to bridge this gap by investigating the technology startup selection process from the perspective of accelerators, utilizing the DEMATEL-ANP-based fuzzy PROMETHEE II approach. To the best of our knowledge, no prior research has scrutinized this hybrid method in evaluating startups. Accordingly, our study explores its feasibility and effectiveness. A ranking method based on spread areas is proposed with formulas to support the decision-making process, and a comparison is conducted to demonstrate the method's advantages. Subsequently, a numerical example is presented to elucidate the complete process of the hybrid method.

The subsequent sections of this paper are structured as follows. Section 2 provides a literature review of the accelerator, selection criteria, and MCDM techniques. Section 3 introduces the classical concept of fuzzy set theory and outlines the hybrid DEMATEL-ANP-based fuzzy PROMETHEE II method. In Section 4, a comparative analysis is presented to underscore the advantages of the ranking technique. Section 5 presents a numerical example that illustrates the applicability and implementation of the hybrid approach in real-world problems. Finally, Section 6 concludes the work by summarizing key findings and suggesting potential avenues for future research.

## 2. Literature Review

### 2.1. Accelerators and the Startup Selection Approach

In the last 15 years, accelerators have boomed due to their effects on startup development, entrepreneurial ecosystem formation, and innovation support [10]. The Y-Combinator, the first accelerator founded by Paul Graham in 2005, was a milestone for the growth of startup accelerators worldwide. By April 2023, according to Seed-DB, 8153 companies were accelerated with funding of USD 88,874,580,633 [11]. Worldwide

high-impact accelerators include Y-Combinator, with 1801 companies accelerated and USD 52,211,811,615 of funding, Techstars with 1336 companies accelerated and USD 12,690,624,018 of funding, and 500 startups with 1686 companies accelerated and USD 4,030,020,819 of funding. In the entrepreneurial ecosystem, many organizations support startups in their early stages with financial and nonfinancial investment, including incubators, accelerators, angel investors, venture capitalists, and governments. However, accelerators are the primary players with their mission of fostering innovative ecosystems and nurturing startups.

Accelerators provide mentoring and networking for selected startups in their intensive programs that develop startups' ability to seek investors. "Accelerators are organizations that serve as gatekeepers and validators of promising business innovations through their embeddedness in their respective ecosystems and, thus, play an active and salient role in socioeconomic and technological advancement" ([10], p. 2). Moreover, various accelerators require equity to counterbalance the support services. For example, the structured investment of one of the biggest accelerators, 500 startups, is USD 150,000 for 6% of their companies [12]. The primary return of profit-driven accelerators is from initial public offerings or acquisitions when a startup exits [13]. Therefore, accelerators must be selective when evaluating startup projects. The filtering process is crucial yet challenging for both accelerators and startups; however, research on the selection criteria and process is still lacking [5].

When investigating the Singapore-based Joyful Frog Digital Incubator (JFDI), Yin and Luo [5] adopted an RWW framework for innovation projects to apply to the accelerator program's assessment. Using a scoreboard of 30 criteria based on the RWW framework, they identified eight vital criteria in the initial screening process. Among these factors, market attractiveness factors explain the existing markets and potential customers, including "demand validation", "customer affordability", and "market demographics", and product feasibility factors include "concept maturity", "sales and distribution", and "product maturity". In addition, product advantage factors, such as "value proposition" and "sustainable advantage", and team competence factors, such as "technology expertise", "prior startup experience", and "feedback mechanism", were crucial. Furthermore, "growth strategy" was considered an essential criterion.

Mariño-Garrido et al. [14] used statistical methods on a Spanish accelerator case study analysis to determine the essential criteria for selecting an entrepreneurial project. Out of the nine criteria investigated, six were significant: speed of acceleration, the extent of innovation, the extent of investment ability, creativity, negotiation, and the extent of team consistency.

Learning about ranking startup methods is crucial for investors, incubators, accelerators, and other stakeholders as it facilitates effective decision-making, risk management, resource allocation, and benchmarking and ultimately increases the chances of success in the dynamic and competitive startup ecosystem. More recent studies about startups can be found at [15–17].

### 2.2. DEMATEL

MCDM methods assist in resolving complex problems that entail multiple objectives, criteria, and alternatives evaluated by decision-makers (DMs). A review of MCDM methods can be found in various studies [18–21].

DEMATEL [22,23] is a constructive method for identifying cause–effect-linked components of a multiplex system. Using a visual systemic model, the technique evaluates interrelationships among criteria and uncovers the critical interrelationships. Moraga et al. [24] used DEMATEL to create a quantitative strategy map identifying causal relationships. Using an MCDM method, the authors developed the final strategy map with qualitative and quantitative approaches that improve and assist managers' assessment process. Altuntas and Gok [25] applied DEMATEL to making correct quarantine decisions, aiming to reduce the burden of the COVID-19 pandemic on the hospitality industry. In 2023,

Wang et al. [26] suggested a new approach for group recommendation, named GroupRecD, which utilizes data mining and the DEMATEL technique to allocate user weights scientifically and rationally. Si et al. [27] conducted a systematic review of DEMATEL. They claimed that the DEMATEL has advantages, including effectively analyzing the direct and indirect effects among factors, visualizing the interdependent relationships between factors by network relation maps, and identifying critical criteria. However, the review also pointed out that DEMATEL cannot achieve the desired level of alternatives, as in Vise Kriterijumska Optimizacija I Kompromisno Resenje (VIKOR) method, or produce partial ranking sequences, as in the ELimination Et Choix Traduisant la REalite (ELECTRE) method. Hence, the DEMATEL was combined with different MCDM methods to obtain appropriate outcomes [27].

### 2.3. AHP

Saaty [28] introduced both the AHP and ANP methods. The AHP method [29] assumes criteria independence and analyzes decision-making problems in a hierarchical criteria structure. To overcome this limitation, Saaty [30,31] developed the ANP method, which considers dependencies and feedback among elements in a network structure to obtain criteria weights. A systematic review of both methods can be found in [32]. The ANP method has been applied to various fields of research. Galankashi et al. [33] amalgamated fuzzy logic and linguistic expression with ANP for investment portfolio selection. When sorting portfolios, multiple studies have focused on financial factors; however, the results indicated that other factors, such as risk, the market, and growth, are essential. The study demonstrated that ANP could present the internal relations between criteria, which is critical in decision-making. In 2023, Saputro et al. [34] utilized Multi-Dimensional Scaling (MDS) and ANP to examine the sustainability approach for developing rural tourism in Panjalu, Ciamis, Indonesia. Kadoić [35] noted that the ANP method effectively analyzes interconnections and consistency within a decision system. When the criteria are interdependent, only the ANP technique can be used [36]. When rating startups, the evaluation values may change over time, thus a network structure to express interdependencies is required, and the original weight of each criterion should be turned into the comprehensive weight. As a result, ANP is chosen over AHP to deal with this problem more effectively.

### 2.4. Fuzzy PROMETHEE II

The Preference Ranking Organization Method for Enrichment Evaluations (PROMETHEE), developed by Brans [37], is one of the most common MCDM methods. PROMETHEE was extended to decision-making in many studies, such as PROMETHEE I for partial ranking and PROMETHEE II for complete ranking [38]. The method has undergone many modifications and improvements to assist humans in decision-making [39]. Among them, PROMETHEE II is the most frequently used because it allows a DM to establish a full ranking [40]. Numerous studies have applied hybrid models combining the PROMETHEE method and other MCDM techniques. Khorasaninejad et al. [41] used a hybrid model to determine the best prime mover in a thermal power plant. The model combined fuzzy ANP-DEMATEL to assess criteria importance and relationships and PROMETHEE to rank alternatives. Govindan et al. [42] used an integrated Fuzzy Delphi, a DEMATEL-based ANP (DANP), and a PROMETHEE method to choose the best supplier based on corporate social responsibility practices and to identify the key factors. Seikh and Mandal [40] proposed an integrated approach, combining PROMETHEE II and SWARA within a fuzzy environment, to streamline the selection of the best bio-chemistry waste management organization. The effectiveness and practicality of their approach were demonstrated through a case study. Hua and Jing [43] extended the classical PROMETHEE method by incorporating the generalized Shapley value in interval-valued Pythagorean fuzzy sets to achieve a more rigorous ranking outcome. To verify the effectiveness of this approach, a case study is conducted to evaluate sustainable suppliers.

The comprehensive literature review conducted in this study revealed the effectiveness and reliability of combining DEMATEL, ANP, and PROMETHEE methods in assisting decision-making in various fields. However, despite the proven success of these individual methods, the amalgamation of DEMATEL-ANP-based fuzzy PROMETHEE II has not been previously applied. Given the intricate nature of rating startups, a robust hybrid approach is essential to effectively address the complexities involved. Considering the multitude of qualitative and quantitative criteria that need to be evaluated, the incorporation of DEMATEL in the initial stage becomes crucial. DEMATEL allows for the examination of cause–effect relationships among these criteria, facilitating the identification and elimination of nonsignificant factors. Subsequently, ANP emerges as the optimal choice for determining criterion weights, as it accounts for criterion interdependencies and provides a comprehensive weighting scheme. To establish a complete ranking, the fuzzy-based PROMETHEE II method is employed with utmost precision. This method accommodates the inherent uncertainties and subjectivity in decision-making processes, enabling a more robust assessment of the startups. By integrating DEMATEL, ANP, and PROMETHEE II within a fuzzy framework, this hybrid approach offers a novel and effective solution for the evaluation and ranking of startups, particularly in contexts where qualitative and quantitative criteria interact and require comprehensive analysis.

### 2.5. Ranking Fuzzy Numbers

Lofi Zadeh [44] introduced fuzzy sets to efficiently model human thought. Fuzzy sets have widely affected many areas of scientific research, including mathematics [45], engineering [46], business, and management [47]. A literature review of the historical evolutions of fuzzy sets, their application, and their frequencies was conducted by Kahraman et al. [48].

Ranking FNs became a critical problem in linguistic decision-making. Jain [49] proposed the first FN ranking method based on maximizing sets. Since then, various methods have been presented, such as the Pos index and its dual Nec index [50], maximizing set and minimizing set [51], area compensation [52], an area method using a radius of gyration [53], deviation degree [54], defuzzified values, heights and spreads [55] and mean of relative values [56].

Wang et al. [54] proposed a ranking method based on left and right deviation degrees derived from maximal and minimal reference sets. Additionally, Wang and Luo [57] introduced an area ranking method using positive and negative ideal points, which they claimed more effectively discriminated FNs than Chen's maximizing and minimizing sets [51]. Asady [58] pointed out that the methods of Wang et al. [54] could not correctly rank fuzzy images. Therefore, he proposed a revised method using parametric forms. Nejad and Mashinchi [59] developed a technique based on the left and right areas to improve the deviation degree method. Yu et al. [60] proposed an extension using an epsilon-deviation degree. Nevertheless, Chutia [61] observed that the approach of Yu et al. still presented limitations in discriminating FNs. Chutia suggested a modified method constituting the ill-defined magnitude value and the angle of the fuzzy set. However, this method cannot be used when FNs have non-linear left and right membership functions [61]. Ghasemi et al. [62] discovered a disadvantage in both the deviation degree method [54] and area ranking based on positive and negative ideal points [57]. The author accordingly introduced an improved approach that considers DMs' risk attitudes. Moreover, numerical examples that demonstrated the efficiency of ranking the proposed method's FNs were provided.

Chu and Nguyen [63] suggested a method to improve Chen's [51] maximizing and minimizing sets to rank FNs. In their study, comparative examples were provided. An experiment demonstrated that the relative maximizing and minimizing set (RMMS) could consistently and logically rank the final fuzzy values of alternatives. This study proposed a fuzzy ranking approach inspired by area ranking and using four spread areas. Based on the RMMS model, the areas were measured and integrated with a confidence level $\mu$ to

assist the FN ranking procedure. The DMs provided confidence levels, which indicated their confidence toward alternatives.

## 3. Model Establishment

### 3.1. Fuzzy Set Theory

*Fuzzy Sets*

$A = \{(x, f_A(x))|x \in U\}$ where $x$ is an element in the space of points $U$, $A$ is a fuzzy set in $U$, $f_A(x)$ is the membership function of $A$ at $x$ [44]. The larger $f_A(x)$, the stronger the grade of membership for $x$ in $A$.

*Fuzzy Numbers*

A real FN $A$ is described as any fuzzy subset of the real line $R$ with a membership function $f_A$ that possesses the following properties [44]. $f_A$ is a continuous mapping from $R$ to [0, 1], $f_A(x) = 0$ for all $x \in (-\infty, a]$. $f_A$ is strictly increasing on the left membership function $[a, b]$ and is strictly decreasing on the right membership function $[c, d]$. $f_A(x) = 1$ for all $x \in [b, c]$ and $f_A(x) = 0$ for all $x \in [d, \infty)$, where $a$, $b$, $c$, and $d$ are real numbers.

We may let $a = -\infty$, or $a = b$, or $b = c$, or $c = d$, or $d = +\infty$. Unless elsewhere defined, A is assumed to be convex, normalized, and bounded, i.e., $-\infty < a, d < \infty$. A can be indicated as $[a, b, c, d]$, $a \le b \le c \le d$. Let $f_A^L(x)$, $a \le x \le b$ represent and $f_A^R(x)$, $c \le x \le d$ represent the left and the right membership function of $A$, respectively, and $f_A(x) = 1$, $b \le x \le c$.

In this research, TFNs will be used. The FN $A$ is a TFN if its membership function $f_A$ is given as follows [51].

$$f_A(x) = \begin{cases} (x - a)/(b - a), & a \le x \le b, \\ (x - c)/(b - c), & b \le x \le c, \\ 0, & \text{otherwise,} \end{cases} \qquad (1)$$

where $a$, $b$, and $c$ are real numbers.

*α-Cuts*

The $\alpha$-cuts of FN $A$ can be determined as $A^\alpha = \{x|f_A(x) \ge \alpha\}$, $\alpha \in [0, 1]$, where $A^\alpha$ is a non-empty bounded closed interval is contained in $R$ and can be denoted by $A^\alpha = [A_l^\alpha, A_u^\alpha]$, where $A_l^\alpha$ are lower bounds and $A_u^\alpha$ are upper bounds [64].

*Arithmetic Operations on Fuzzy Numbers*

Given FNs $A$ and $B$, $A$, $B \in R^+$, $A^\alpha = [A_l^\alpha, A_u^\alpha]$ and $B^\alpha = [B_l^\alpha, B_u^\alpha]$. By the interval arithmetic, some primary operations of $A$ and $B$ can be described as follows [64].

$$(A \oplus B)^\alpha = [A_l^\alpha + B_l^\alpha, \ A_u^\alpha + B_u^\alpha] \qquad (2)$$

$$(A \ominus B)^\alpha = [A_l^\alpha - B_u^\alpha, A_u^\alpha - B_l^\alpha] \qquad (3)$$

$$(A \otimes B)^\alpha = [A_l^\alpha \cdot B_l^\alpha, \ A_u^\alpha \cdot B_u^\alpha] \qquad (4)$$

$$r(A)^\alpha = [r \cdot A_l^\alpha, \ r \cdot A_u^\alpha], \ r \in R^+ \qquad (5)$$

*Linguistic Values*

A linguistic variable is a variable whose values are represented in linguistic terms. It is advantageous for dealing with complicated matters or is ambiguous to be rationally described in traditional quantitative information [51,65]. DMs are assumed to have agreed to weight alternatives over criteria using linguistic values such as *Extremely Poor (EP)*, *Very Poor (VP)*, *Poor (P)*, *Moderate (M)*, *High (H)*, *Very High (VH)*, and *Extremely High (EH)* which can

also be represented by TFNs such as *EP* = (0,0.1,0.25), *VP* = (0.1,0.2,0.35), *P* = (0.25,0.35,0.5), *M* = (0.35,0.5,0.65), *H* = (0.5,0.65,0.75), *VH* = (0.65,0.8,0.9), and *EH* = (0.75,0.9,1).

### 3.2. Relative Maximizing and Minimizing Sets

Chu and Nguyen [63] suggested a technique to improve Chen's [51] maximizing and minimizing set to rank FNs. In their study, numerical comparisons and examples were conducted to demonstrate that the RMMS can consistently and logically rank fuzzy values of alternatives. The RMMS [63] technique is introduced as follows.

Assume there are $n$ FNs $A_i = (a_i, b_i, c_i)$, $i = 1, \ldots, n$, $n \geq 2$, $f_{A_i} \in R$. $x_{\min} = \inf S$, $x_{\max} = \sup S$, $S = U_{i=1}^n S_i$, $S_i = \{x | f_{A_i}(x) > 0\}$. FNs $A_g = (a_g, b_g, c_g)$ and $A_l = (a_l, b_l, c_l)$ are added to the right and left sides of the above $n$ FNs $A_i = (a_i, b_i, c_i)$, $i = 1, \ldots, n$, respectively. Assume $x_{\min} = a_1$, $x_{\max} = c_n$, $c_g \geq x_{\max}$ and $a_l \leq x_{\min}$. Let $\delta_R = c_g - x_{\max}$ and $\delta_L = x_{\min} - a_l$, where $x_{\max} = c_n$, $x_{\min} = a_1$, $\delta_R \geq 0$, $\delta_L \geq 0$. The new supremum element is defined as $x'_{\max} = x_{\max} + \delta$ and the new infimum element is defined as $x'_{\min} = x_{\min} - \delta$, where $\delta = \max\{\delta_L, \delta_R\}$.

The relative maximizing set $M'$ and the relative minimizing set $N'$ are determined as:

$$f_{M'}(x) = \begin{cases} \left( \frac{x_{R_i} - (x_{\min} - \delta)}{(x_{\max} + \delta) - (x_{\min} - \delta)} \right)^k & , (x_{\min} - \delta) \leq x_{R_i} \leq (x_{\max} + \delta) \\ 0, \text{ otherwise} \end{cases} \qquad (6)$$

$$f_{N'}(x) = \begin{cases} \left( \frac{x_{L_i} - (x_{\max} + \delta)}{(x_{\min} - \delta) - (x_{\max} + \delta)} \right)^k & , (x_{\min} - \delta) \leq x_{L_i} \leq (x_{\max} + \delta) \\ 0, \text{ otherwise} \end{cases} \qquad (7)$$

Herein, $k$ is set to 1. The value of $k$ can be varied to suit the application. The total relative utility of each $A_i$ is denoted as in Equation (8).

$$U_{T'}(A_i) = \frac{1}{4}[U_{R_{i1}}(A_i) + ((1 - U_{L_{i1}}(A_i)) + U_{L_{i2}}(A_i) + ((1 - U_{R_{i2}}(A_i))], i = 1, \ldots, (n + 2) \qquad (8)$$

where the first right relative utility $U_{R_{i1}}(A_i) = \sup\left( f_{M'}(x) \wedge f_{A_i}^R(x) \right)$, the first left relative utility $U_{L_{i1}}(A_i) = \sup\left( f_{N'}(x) \wedge f_{A_i}^L(x) \right)$, the second left relative utility $U_{L_{i2}}(A_i) = \sup\left( f_{M'}(x) \wedge f_{A_i}^L(x) \right)$ and the second right relative utility $U_{R_{i2}}(A_i) = \sup\left( f_{N'}(x) \wedge f_{A_i}^R(x) \right)$.

### 3.3. Spread Area-Based RMMS

In 2011, Nejad and Mashinchi [59] pointed out the shortcomings of Wang et al.'s [54] deviation degree method that when the values of the left area, the right area, the transfer coefficient $\lambda_i$ or $1 - \lambda_i$ is zero, the ranking result is inaccurate. Hence, to prevent these problems from occurring, expanding $x_{\max}$ and $x_{\min}$ is needed when ranking. Chu and Nguyen [63] also found out that when adding a new FN, $x_{\max}$ and $x_{\min}$ must be modified by adding equal values to consider both sides of membership functions. Consequently, four utilities need to be accounted for to reduce the inconsistency of Chen's [51] maximizing and minimizing set. However, if a set of FNs with $x_{\min} = 3$, then a new FN $A_g = (3, 3, 3)$ is added, there is no extended value applicable in this situation. Therefore, this work suggests integrating confidence levels in ranking FNs to solve the mentioned problems.

Yeh and Kuo [66] in their research on evaluating passenger service quality of Asia-Pacific international airports, suggested incorporating a DM's confidence level $\alpha$ and a preference index $\lambda$ to obtain an overall service performance index. In the evaluation procedure, DMs give the value $\alpha$, based on the concept of an $\alpha$-cut, with respect to the criteria's weights and alternative performance ratings.

This work proposes to use confidence level in a new perspective, which is confidence level, symbolized as $\mu$, will be integrated into measuring areas spreading based on the RMMS model to assist the ranking FNs procedure, as shown in Figure 1. First, $h$ experts

in the group of DMs, $D = \{D_1, \ldots, D_e, \ldots, D_h\}$ are asked to specify their confidence $\mu_{D_e}$, representing their confidence for alternatives to obtain $\mu = \frac{\sum_e^h \mu_{D_e}}{h}$, $\mu_{D_e} \in [0, 1]$. The greater the $\mu$, the more assured is the decision-maker on the alternative.

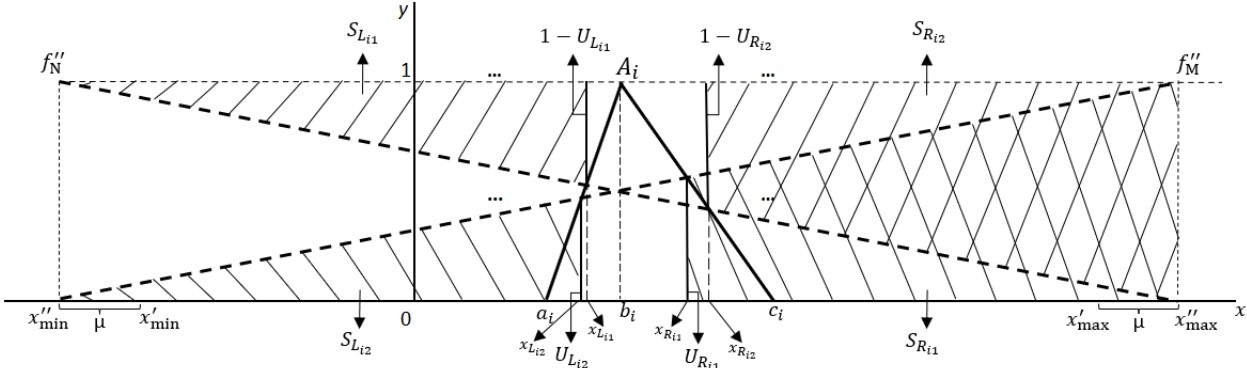

**Figure 1.** Spread area-based RMMS ranking method.

Since DMs' confidence in an alternative will influence their confidence level in other alternatives, the confidence level $\mu$, calculated by the average of all DMs' evaluation, should be engaged simultaneously with the immensity of the RMMS concept. Accordingly, value $\mu$ is integrated by shifting the RMMS's infimum element to the left, provided that the new infimum element is obtained as $x''_{min} = x'_{min} - \mu$. Similarly, the average value of $\mu$ will be integrated by shifting the RMMS's infimum element to the right, provided that the new supremum element is obtained as $x''_{max} = x'_{max} + \mu$.

The coordinates of the intersection of the $A_i$ with the relative maximizing set $M''$ and the relative minimizing set $N''$ can be seen in Figure 1 and are determined as the following equations.

$$x_{L_{i1}} = \frac{bx''_{max} - ax''_{min}}{b - a - x''_{min} + x''_{max}} \tag{9}$$

$$x_{L_{i2}} = \frac{bx''_{min} - ax''_{max}}{x''_{min} - x''_{max} + b - a} \tag{10}$$

$$x_{R_{i1}} = \frac{bx''_{min} - cx''_{max}}{b - c + x''_{min} - x''_{max}} \tag{11}$$

$$x_{R_{i2}} = \frac{cx''_{min} - bx''_{max}}{c - b + x''_{min} - x''_{max}} \tag{12}$$

The first left spread area $S_{L_{i1}}$ is defined as follows.

$$
\begin{aligned}
S_{L_{i1}}(A_i) &= \int_{x''_{min}}^{x_{L_{i1}}} 1 dx - \int_{x''_{min}}^{x_{L_{i1}}} f''_N(x)dx \\
S_{L_{i1}}(A_i) &= x_{L_{i1}} - x''_{min} - \int_{x''_{min}}^{x_{L_{i1}}} \left( \frac{x - x''_{max}}{x''_{min} - x''_{max}} \right) dx \\
&= x_{L_{i1}} - x''_{min} - \left( \frac{x^2}{2(x''_{min} - x''_{max})} - \frac{xx''_{max}}{x''_{min} - x''_{max}} \right) \Big|_{x'''_{min}}^{x_{L_{i1}}} \\
&= x_{L_{i1}} - x''_{min} - \left( \frac{x_{L_{i1}}^2 - 2x_{L_{i1}}x''_{max}}{2(x''_{min} - x''_{max})} - \frac{x''_{min}{}^2 - 2x''_{min}x''_{max}}{2(x''_{min} - x''_{max})} \right) \\
&= \frac{(x_{L_{i1}} - x''_{min})(2x''_{min} - 2x''_{max} - x_{L_{i1}} - x''_{min} - 2x''_{max})}{2(x''_{min} - x''_{max})} = \frac{(x_{L_{i1}} - x''_{min})(x''_{min} - 4x''_{max} - x_{L_{i1}})}{2(x''_{min} - x''_{max})}
\end{aligned} \tag{13}
$$

If the first left spread area $S_{L_{i1}}$ is larger, the fuzzy number $A_i$ is larger. The second left spread area $S_{L_{i2}}$ is defined as Equation (14); and if $S_{L_{i2}}$ is larger, the fuzzy number

$A_i$ is also larger. The first right spread area $S_{R_{i1}}$ is defined as Equation (15); but if $S_{R_{i1}}$ is larger, the fuzzy number $A_i$ is smaller. Finally, the second right spread area $S_{R_{i2}}$ is defined as Equation (16); and if $S_{R_{i2}}$ is larger, the fuzzy number $A_i$ is also smaller. Therefore, the above four areas must be considered when ranking FNs. The detailed derivation for Equations (14)–(16) is placed in Appendix A.

$$S_{L_{i2}}(A_i) = \int_{x''_{\min}}^{x_{L_{i2}}} f''_M(x)dx = \frac{(x_{L_{i2}} - x''_{\min})^2}{2(x''_{\max} - x''_{\min})} \tag{14}$$

$$S_{R_{i1}}(A_i) = \int_{x_{R_{i1}}}^{x''_{\max}} f''_M(x)dx = \frac{(x''_{\max} + x_{R_{i1}})(x''_{\max} - x_{R_{i1}} - 2x''_{\min})}{2(x''_{\max} - x''_{\min})} \tag{15}$$

$$S_{R_{i2}}(A_i) = \int_{x''_{R_{i2}}}^{x''_{\max}} 1dx - \int_{x''_{R_{i2}}}^{x''_{\max}} f''_N(x)dx = \frac{(x''_{\max} - x_{R_{i2}})(2x''_{\min} - x''_{\max} + x_{R_{i2}})}{2(x''_{\min} - x''_{\max})} \tag{16}$$

Finally, the ranking value of each $A_i$ is determined as Equation (17) to classify FNs. An FN is more prominent if its value is larger.

$$V(A_i) = \frac{1}{4}\left(S_{L_1}(A_i) - S_{R_1}(A_i) + S_{L_2}(A_i) - S_{R_2}(A_i)\right) \tag{17}$$

*3.4. The Hybrid DEMATEL-ANP Based Fuzzy PROMETHEE II Model*

3.4.1. DEMATEL

The DEMATEL method is first used to demonstrate the interrelationships between criteria and produce the influential network relationship map. The constructing equations of the classical DEMATEL model can be summarized as follows [67].

Assume that $h$ experts in a decision group $D = \{D_1, D_2, \ldots, D_h\}$ are asked to indicate the direct effect of factor (criterion) $C_i$ has on factor (criterion) $C_j$ in a system with $m$ factors (criteria) $C = \{C_1, C_2, \ldots, C_m\}$ using an integer scale of No Effect (0), Low Effect (1), Medium Low Effect (2), Medium Effect (3), Medium High Effect (4), High Effect (5) and Extremely Strong Effect (6). Next, the individual direct-influence matrix $Z_e = \left[z_{ij}^e\right]_{m \times m}$ provided by the $e$th expert can be constructed, where all main diagonal components are equal to zero and $z_{ij}^e$ represent the respondent's evaluation of DM on the degree to which criterion $C_i$ affects $C_j$.

Step 1. Generating the group direct-influence matrix. By aggregating $h$ DMs' judgments, the group direct-influence matrix $Z = \left[z_{ij}\right]_{m \times m}$ can be constructed by

$$Z = \frac{1}{h}\sum_{e=1}^{h} z_{ij}, i, j = 1, 2, \ldots, m. \tag{18}$$

Step 2. Acquiring the normalized direct-influence matrix. At this step, the normalized direct-influence matrix by the $e$th expert is $X_e = \left[x_{ij}^e\right]_{m \times m}, e = 1, 2, \ldots h$.

The following equations calculate the average matrix X

$$X = \frac{(x^1 \oplus x^2 \oplus \ldots \oplus x^h)}{h} \tag{19}$$

$$x_{ij} = \frac{\sum_{e=1}^{h} x_{ij}^e}{h} \tag{20}$$

Step 3. Computing the total-influence matrix T. The total-influence matrix $T = \left[t_{ij}\right]_{m \times m}$ is computed as the summation of the direct impacts and all the indirect impacts by Equation (21)

$$T = X + X^2 + X^3 + \ldots + X^h = X(I - X)^{-1}, \tag{21}$$

when $h \to \infty$ in the identity matrix, known as *I*.

Step 4. Setting up a threshold value and producing the causal diagram.

The sum of columns and the sum of rows are symbolized as *R* and *D*, respectively, within the total-relation matrix $T = [t_{ij}], \{i, j \in 1, 2, \ldots, m\}$ by the following formulas:

$$D = [d_i]_{m \times 1} = \left[ \sum_{j=1}^{m} t_{ij} \right]_{m \times 1} \tag{22}$$

$$R = [r_j]_{1 \times m} = \left[ \sum_{i=1}^{m} t_{ij} \right]_{1 \times m} \tag{23}$$

The horizontal axis vector (*D* + *R*) called "Prominence" demonstrates the power of influence degree that is given and received by the criteria. The vertical axis vector (*D* − *R*) named "Relation" shows the system's criteria effect. If (*D* − *R*) is positive, the criterion $C_j$ influences other criteria and can be grouped into a causal group; if (*D* + *R*) is negative, the criterion $C_j$ is being influenced by the other criteria and can be grouped into an effect group. A causal diagram can be produced by mapping the (*D* + *R*, *D* − *R*) dataset, yielding valuable assessment perception. A threshold value can be defined to screen out the negligible factors [68,69]. In this work, factors that have a value higher than the average value of the "Prominence" (*D* + *R*) and/or (*D* − *R*) is positive are selected to use in the next step.

### 3.4.2. ANP

Next, the present work applied the ANP method to produce the weights of the criteria. The generalized ANP process from previous studies is summarized as follows [30,70,71]. In this work, a set of importance scales [13] is adopted to weight each criterion using linguistic values, including 1—*Identically Important (II)*, 3—*Moderately Important (MI)*, 5—*Highly Important (HI)*, 7—*Very Highly Importance (VHI)*, 9—*Extremely Important (EI)*, and 2, 4, 6, 8 are the median values. Reciprocal values are used for inverse comparison.

Step 1. Obtaining Pairwise Comparison Matrix (PCM). Assume that h experts in a decision group $D = \{D_1, D_2, \ldots, D_h\}$ are responsible for evaluating criteria $C = \{C_1, C_2, \ldots, C_m\}$ that are screened through the previous step. The PCM is generated by comparing the *i*th row with the *j*th column. The weights of components are formed as shown in matrix *A*. The diagonal components with identical importance are illustrated by 1.

$$A = [a_{ij}]_{m \times m} \begin{bmatrix} 1 & a_{12} & \cdots & a_{1m} \\ a_{21} & 1 & & \\ \vdots & & 1 & \\ a_{m1} & & & 1 \end{bmatrix}$$

As there are several DMs, the pairwise comparison values from different DMs may vary. Experts can decide together, or each assessment can be integrated into a PCM by the geometric mean *GM* as in Equation (24).

$$GM = \sqrt[j]{i_1 i_2 i_3 \ldots i_j} \tag{24}$$

Step 2. Computing eigenvectors and the unweighted supermatrix. In this step, eigenvector $E_i$ is obtained through Equation (25), which is computed by each row's average.

$$E_i = \frac{1}{m} \sum_{j=1}^{m} a_{ij} \tag{25}$$

Then, the eigenvectors of each matrix are consolidated to form the unweighted matrix.

Step 3. Examining the consistency. To guarantee consistency among the judgments of the DMs, it is necessary to test the consistency by three metrics, including Consistency Measure (*CM*), Consistency Index (*CI*), and Consistency Ratio (*CR*).

The general form for *CM* values is obtained through Equation (26).

$$CM_j = \frac{a_j \times E}{E_j}, \tag{26}$$

where $j = 1,2,3, \ldots, m$, $a_j$ is the corresponding row of the comparison matrix, $E$ is Eigenvector and $E_j$ represents the corresponding component in $E$.

Then, $\lambda_{\max}$ is obtained by the average of the *CM* vector. The *CI* is calculated as shown in Equation (27).

$$CI = \frac{\lambda_{\max} - m}{m - 1} \tag{27}$$

Next, a random index, as listed in Table 1 [13], is computed following the order of the PCM. Consequently, the Consistency Ratio *CR* is obtained by Equation (28).

$$CR = \frac{CI}{RI} \tag{28}$$

**Table 1.** Random Index.

| Order | 1 | 2 | 3 | 4 | 5 | 6 | 7 | 8 | 9 | 10 |
|-------|---|---|------|------|------|------|------|------|------|------|
| R.I   | 0 | 0 | 0.52 | 0.89 | 1.11 | 1.25 | 1.35 | 1.40 | 1.45 | 1.49 |

The value of $CR \leq 0.1$ is in the satisfactory range; otherwise, the pairwise comparison is required to be revised.

Step 4. Obtain the weighted supermatrix. A weighted supermatrix is obtained to evaluate the relation between criteria. Then, the unweighted matrix is converted into a weighted supermatrix to make the sum of each column 1, called column stochastic.

Step 5. Determining stable weights by obtaining limit supermatrix. The values produced from the previous step are elevated to the power of $2k$ until the values are firmly established, where $k$ is an arbitrarily large number. The final priorities can be determined using the normalization function on each block of the limit matrix. The most significant value represents the most critical criterion among other criteria. The stable weights $w$ constructed from this step are utilized in the following steps.

### 3.4.3. Fuzzy PROMETHEE-Based Ranking Method

The same group of $h$ experts $D = \{D_1, D_2, \ldots, D_h\}$ will assess $n$ alternatives $A = \{A_1, A_2, \ldots, A_n\}$ under $m$ criteria $C = \{C_1, C_2, \ldots, C_m\}$ that are screened through the previous steps. Let $f_{ij}^e = (a_{ij}^e, b_{ij}^e, c_{ij}^e)$, $i = 1, 2 \ldots, n$, $j = 1, 2 \ldots, m$, $e = 1, 2 \ldots, h$, be the rating assigned to an alternative $A_i$ under the criterion $C_j$ by a decision-maker $D_e$. Criteria chosen from the earlier steps are first categorized into the cost-benefit framework as qualitative benefit criteria, $C_j$, $j = 1, 2 \ldots, k$, quantitative benefit criteria, $C_j$, $j = k + 1, \ldots, k'$, cost qualitative criteria, $C_j$, $j = k' + 1, \ldots, k''$, and cost quantitative criteria $C_j$, $j = k'' + 1, \ldots, m$. The fuzzy PROMETHEE II process is summarized as follows [72,73].

Step 1. Constructing the fuzzy decision matrix. Aggregated rating $f_{ij} = (a_{ij}, b_{ij}, c_{ij})$ is:

$$f_{ij} = \left(\frac{1}{h}\right) \otimes \left(f_{ij1} \oplus \ldots \oplus f_{ije} \oplus \ldots \oplus f_{ijh}\right) \tag{29}$$

where $a_{ij} = \sum\limits_{e=1}^{h} \frac{a_{ije}}{h}$, $b_{ij} = \sum\limits_{e=1}^{h} \frac{b_{ije}}{h}$, $c_{ij} = \sum\limits_{e=1}^{h} \frac{c_{ije}}{h}$.

Step 2. Computing the normalized matrix. The normalization is completed using the Chu and Nguyen [63] approach. The ranges of normalized TFNs belong to [0, 1]. Suppose $l_{ij} = (al_{ij}, bl_{ij}, cl_{ij})$ is the value of an alternative $A_i$, $i = 1, 2, \ldots, n$, versus a benefit (B) criterion or a cost (C) criterion. The normalized value $l_{ij}$ can be as

$$l_{ij} = \left( \frac{al_{ij} - al_j^*}{y_j^*}, \frac{bl_{ij} - al_j^*}{y_j^*}, \frac{cl_{ij} - al_j^*}{y_j^*} \right), j \in \text{B}, \tag{30}$$

$$l_{ij} = \left( \frac{cl_j^* - cl_{ij}}{y_j^*}, \frac{cl_j^* - bl_{ij}}{y_j^*}, \frac{cl_j^* - al_{ij}}{y_j^*} \right), j \in \text{C}, \tag{31}$$

where $al_j^* = \min_i al_{ij}$, $cl_j^* = \max_i cl_{ij}$, $y_j^* = cl_j^* - al_j^*$, $i = 1, 2, \ldots, n$, $j = k' + 1, \ldots, k''$ and $j = k'' + 1, \ldots, m$, $l_{ij} = (al_{ij}, bl_{ij}, cl_{ij})$.

Step 3. Calculating the evaluative differences. Pairwise comparison is made by calculating the evaluative differences of $i$th alternative with respect to other alternatives. The intensity of the fuzzy preference $P_j(A_i, A_{i'})$ of an alternative $A_i$ over $A_{i'}$ is obtained by Equations (32) and (33), based on Equation (3)

$$P'_j\big(C_j(A_i) - C_j(A_{i'})\big) = P'_j(A_i, A_{i'}) \tag{32}$$

$$= l_{ij} - l_{ij'} = (al_{ij}, bl_{ij}, cl_{ij}) - (al'_{ij}, bl'_{ij}, cl'_{ij}) = (al_{ij} - cl'_{ij}, bl_{ij} - bl'_{ij}, cl_{ij} - al'_{ij}) \tag{33}$$

where $P_j$ is the fuzzy preference function for the $j$th criterion and $C_j(A_i)$ is the evaluation of alternative $A_i$ corresponding to criterion $C_j$.

Step 4. Determining the preference function. To avoid the complexity and be in a more practicable form, the simplified fuzzy preference function is applied in this study as in Equations (34) and (35).

$$P'_j(A_i, A_{i'}) = 0 \text{ if } C_j(A_i) \leq C_j(A_{i'}) \tag{34}$$

$$P'_j(A_i, A_{i'}) = (C_j(A_i) - C_j(A_{i'})) \text{ if } C_j(A_i) > C_j(A_{i'}) \tag{35}$$

Step 5. Reckoning the aggregated fuzzy preference function. Calculate the aggregated fuzzy preference function considering the criteria weights computed from the ANP method.

$$\pi'(A_i, A_{i'}) = \sum_{j=1}^{m} w_j P'_j(A_i, A_{i'}) / \sum_{j=1}^{m} w_j \tag{36}$$

The higher $\pi'(A_i, A_{i'})$ is, the stronger preference for the $i$th alternative will be.

Step 6. Determining the fuzzy leaving flow $\varphi'^+(A_i)$ and the fuzzy entering flow $\varphi'^-(A_i)$

The fuzzy leaving flow of $A_i$ is determined as

$$\varphi'^+(A_i) = \frac{1}{n-1} \sum_{\substack{i'=1 \\ i' \neq 1}}^{n} \pi'(A_i, A_{i'}) \tag{37}$$

The fuzzy entering flow of $A_i$ is determined as

$$\varphi'^-(A_i) = \frac{1}{n-1} \sum_{\substack{i'=1 \\ i' \neq 1}}^{n} \pi'(A_{i'}, A_i) \tag{38}$$

Step 7. Calculating the fuzzy net outranking flow for each alternative

$$\varphi'(A_i) = \varphi'^+(A_i) - \varphi'^-(A_i) \tag{39}$$

Step 8. Defuzzifying the fuzzy net outranking flow value and obtaining the ranking of alternatives. In this step, the spread area-based RMMS model is proposed to apply to assist defuzzification and obtain the final ranking using Equations (12)–(20). An FN is more prominent if its value $V(A_i)$ is more significant.

## 4. Numerical Comparison and Consistency Test

In this section, various examples of comparisons are established to investigate the effectiveness of the proposed method. The first example illustrates the ranking orders of the method compared with the methods of Wang et al. [54] and Nejad and Mashinchi [59]. We used FNs in Examples 2–4 from Nejad and Mashinchi [59], and then different situations were generated through the addition of new FNs for testing the consistency of the ranking results, as shown in Table 2. In Situation (1), methods from both Nejad and Mashinchi and Wang et al. produce $A_1 = A_2 \prec A_3$, but the proposed method can discriminate between three FNs with the order $A_1 \prec A_2 \prec A_3$. Furthermore, the ranking order is $A_1 \prec A_2 \prec A_3$, and either $A_4 = (-3, -2, -1)$ is added (see Situation (1.1)) or $A_4 = (8.75, 9.5, 11)$ is added (see Situation (1.2)). In Situation (2), the proposed method yields the same ranking, $A_1 \prec A_2$, as that of the method of Nejad and Mashinchi when either $A_4 = (-1.5, -0.8, -0.6)$ or $A_4 = (1.15, 2.5, 3.15)$ is added. However, the method of Wang et al. highlights the inconsistency and produces $A_1 = A_2$ in Situation (2.2). In Situation (3), the proposed method yields the same ranking $A_1 \succ A_2$ as that of Nejad and Mashinchi when $A_4 = (-5, -4, -3, -1)$ or $A_4 = (6, 6, 7, 8)$ is added, but the method of Wang et al. compensates for the inconsistency and produces $A_1 = A_2$ in Situation (3.2). The first comparison demonstrates the usefulness of the proposed method in discriminating FNs.

**Table 2.** Modified comparison based on Examples 2, 3, and 4 from Nejad and Mashinchi [59].

| Situations | Methods | Results | Results after Adding New FNs | |
|---|---|---|---|---|
| (1) | | | (1.1) $A_4 = (-3, -2, -1)$ | (1.2) $A_4 = (8.75, 9.5, 11)$ |
| $A_1 = (2, 3, 5, 6)$ $A_2 = (1, 4, 7)$ $A_3 = (4, 5, 7)$ | [54] | $A_1 = A_2 \prec A_3$ | $A_2 \prec A_1 \prec A_3$ | $A_1 = A_2 \prec A_3$ |
| | [59] | $A_1 = A_2 \prec A_3$ | $A_1 = A_2 \prec A_3$ | $A_1 = A_2 \prec A_3$ |
| | Proposed method | $A_1 \prec A_2 \prec A_3$ | $A_1 \prec A_2 \prec A_3$ | $A_1 \prec A_2 \prec A_3$ |
| (2) | | | (2.1) $A_4 = (-1.5, -0.8, -0.6)$ | (2.2) $A_4 = (1.15, 2.5, 3.15)$ |
| $A_1 = (0.2, 0.5, 0.8)$ $A_2 = (0.4, 0.5, 0.6)$ | [54] | $A_1 \prec A_2$ | $A_1 \prec A_2$ | $A_1 = A_2$ |
| | [59] | $A_1 \prec A_2$ | $A_1 \prec A_2$ | $A_1 \prec A_2$ |
| | Proposed method | $A_1 \prec A_2$ | $A_1 \prec A_2$ | $A_1 \prec A_2$ |
| (3) | | $A_1 \succ A_2$ | (3.1) $A_4 = (-5, -4, -3, -1)$ | (3.2) $A_4 = (6, 6, 7, 8)$ |
| $A_1 = (1, 2, 5)$ $A_2 = (1, 2, 2, 4)$ | [54] | $A_1 \succ A_2$ | $A_1 \succ A_2$ | $A_1 = A_2$ |
| | [59] | | $A_1 \succ A_2$ | $A_1 \succ A_2$ |
| | Proposed method | $A_1 \succ A_2$ | $A_1 \succ A_2$ | $A_1 \succ A_2$ |

Second, three sets of FNs are created to further examine the proposed method's stability and credibility, as shown in Table 3. In all previous situations, the method of Wang et al. is ineffective in distinguishing FNs. For example, in Situation (1.1), the method of Nejad and Mashinchi yields an FN ranking, $A_1 \prec A_2 \prec A_3 \prec A_4$, but yields $A_1 = A_2 = A_3 = A_4$ in cases (1) and (1.2), indicating inconsistency, but the proposed method yields $A_1 \prec A_2 \prec A_3 \prec A_4$ in all Situations (1), (1.1), and (1.2). Similarly, in Situations (2) and (2.2), the ranking order obtained using the method of Nejad and Mashinchi is $A_1 = A_2 = A_3$; however, when $A_4 = (-7, -5, -3, -2)$ is added, the order changes to $A_1 \prec A_2 \prec A_3$, as in Situation (2.1); whereas the suggested method persistently ranks in the following order: $A_1 \prec A_2 \prec A_3$. In Situations (3) and (3.2), both the proposed method and the method of Nejad and Mashinchi yield a ranking order of $A_1 \prec A_2$; however, in (3.1), when $A_3 = (-4, -2.5, -1.5)$ is added, the method of Nejad and Mashinchi yields $A_1 \succ A_2$; however, the proposed method yields a persistent rank order of $A_1 \prec A_2$. Hence, the second comparison has demonstrated the effectiveness of the proposed method in discriminating FNs compared to Wang et al.'s technique and the consistency compared with the method of Nejad and Mashinchi.

**Table 3.** Comparison with Wang et al.'s [54] and Nejad and Mashinchi [59].

| Situations | Methods | Results | Results after Adding New FNs | |
|---|---|---|---|---|
| (1) | | | (1.1) $A_5 = (-5, -4, -3)$ | (1.2) $A_5 = (8, 9, 10)$ |
| $A_1 = (3, 3, 3)$ | [54] | $A_1 = A_2 = A_3 = A_4$ | $A_1 = A_2 = A_3 = A_4$ | $A_1 = A_2 = A_3 = A_4$ |
| $A_2 = (3, 3, 6)$ $A_3 = (3, 3, 8)$ | [59] | $A_1 = A_2 = A_3 = A_4$ | $A_1 \prec A_2 \prec A_3 \prec A_4$ | $A_1 = A_2 = A_3 = A_4$ |
| $A_4 = (3, 3, 6, 8)$ | Proposed method | $A_1 \prec A_2 \prec A_3 \prec A_4$ | $A_1 \prec A_2 \prec A_3 \prec A_4$ | $A_1 \prec A_2 \prec A_3 \prec A_4$ |
| (2) | | | (2.1) $A_4 = (-7, -5, -3, -2)$ | (2.2) $A_4 = (7, 9, 11, 12)$ |
| $A_1 = (3, 3, 3)$ | [54] | $A_1 = A_2 = A_3$ | $A_1 = A_2 = A_3$ | $A_1 = A_2 = A_3$ |
| $A_2 = (3, 3, 6)$ $A_3 = (3, 3, 5, 6)$ | [59] | $A_1 = A_2 = A_3$ | $A_1 \prec A_2 \prec A_3$ | $A_1 = A_2 = A_3$ |
| | Proposed method | $A_1 \prec A_2 \prec A_3$ | $A_1 \prec A_2 \prec A_3$ | $A_1 \prec A_2 \prec A_3$ |
| (3) | | | (3.1) $A_3 = (-4, -2.5, -1.5)$ | (3.2) $A_3 = (6, 7.8, 8.5)$ |
| $A_1 = (2, 2, 7)$ | [54] | $A_1 = A_2$ | $A_1 = A_2$ | $A_1 = A_2$ |
| $A_2 = (2, 4, 4)$ | [59] | $A_1 \prec A_2$ | $A_1 \succ A_2$ | $A_1 \prec A_2$ |
| | Proposed method | $A_1 \prec A_2$ | $A_1 \prec A_2$ | $A_1 \prec A_2$ |

Additionally, a consistency test is designed to examine the reliability of the proposed method, as shown in Tables 4 and 5. In Example 1, the result is $A_1 \prec A_2 \prec A_3$, $A_1 \prec A_2 \prec A_3$ for all assumed various $\mu$ values. In Example 2, when $A_4 = (8, 9, 10)$ is added, the classifying order remains the same as $A_1 \prec A_2 \prec A_3$ for all $0.1 \prec \mu \prec 1$. Finally, in Example 3, when $A_4 = (-3, -2, -1)$ is added, the proposed method consistently yields an order of $A_1 \prec A_2 \prec A_3$ for all tested values of $\mu$. The results of the numerical comparison demonstrate the credibility and effectiveness of the suggested ranking method based on spread area-based RMMS.

**Table 4.** Numerical comparison with Chu and Nguyen [63].

| Situations | Methods | Results | Results after Adding New FNs | |
|---|---|---|---|---|
| (1) | | | (1.1) $A_3 = (1,4,5)$ | (1.2) $A_3 = (-3,-2,-1)$ |
| $A_1 = (1,3,5)$ $A_2 = (2,3,4)$ | [63] | $A_1 = A_2$ | $A_1 = A_2$ | $A_1 = A_2$ |
| | Proposed method | $A_1 \prec A_2$ | $A_1 \prec A_2$ | $A_1 \prec A_2$ |
| (2) | | | (2.1) $A_3 = (2,3,7)$ | (2.2) $A_3 = (-4,-2,-2)$ |
| $A_1 = (2,2,4)$ $A_2 = (2,2,6)$ | [63] | $A_1 = A_2$ | $A_1 = A_2$ | $A_1 = A_2$ |
| | Proposed method | $A_1 \prec A_2$ | $A_1 \prec A_2$ | $A_1 \prec A_2$ |

**Table 5.** A consistency test with various values of μ in different examples.

| μ | **Examples** | | |
|---|---|---|---|
| | **(1) Three FNs** $A_1 = (2,3,5,6)$, $A_2 = (1,4,7)$ $A_3 = (4,5,7)$ | **(2) Add an FN to the Right Side** $A_1 = (2,3,5,6)$, $A_2 = (1,4,7)$ $A_3 = (4,5,7)$, $A_4 = (8,9,10)$ | **(3) Add an FN to the Left Side** $A_1 = (2,3,5,6)$, $A_2 = (1,4,7)$ $A_3 = (4,5,7)$, $A_4 = (-3,-2,-1)$ |
| 0.1 | $A_1 \prec A_2 \prec A_3$ | $A_1 \prec A_2 \prec A_3$ | $A_1 \prec A_2 \prec A_3$ |
| 0.2 | $A_1 \prec A_2 \prec A_3$ | $A_1 \prec A_2 \prec A_3$ | $A_1 \prec A_2 \prec A_3$ |
| 0.3 | $A_1 \prec A_2 \prec A_3$ | $A_1 \prec A_2 \prec A_3$ | $A_1 \prec A_2 \prec A_3$ |
| 0.4 | $A_1 \prec A_2 \prec A_3$ | $A_1 \prec A_2 \prec A_3$ | $A_1 \prec A_2 \prec A_3$ |
| 0.5 | $A_1 \prec A_2 \prec A_3$ | $A_1 \prec A_2 \prec A_3$ | $A_1 \prec A_2 \prec A_3$ |
| 0.6 | $A_1 \prec A_2 \prec A_3$ | $A_1 \prec A_2 \prec A_3$ | $A_1 \prec A_2 \prec A_3$ |
| 0.7 | $A_1 \prec A_2 \prec A_3$ | $A_1 \prec A_2 \prec A_3$ | $A_1 \prec A_2 \prec A_3$ |
| 0.8 | $A_1 \prec A_2 \prec A_3$ | $A_1 \prec A_2 \prec A_3$ | $A_1 \prec A_2 \prec A_3$ |
| 0.9 | $A_1 \prec A_2 \prec A_3$ | $A_1 \prec A_2 \prec A_3$ | $A_1 \prec A_2 \prec A_3$ |
| 1.0 | $A_1 \prec A_2 \prec A_3$ | $A_1 \prec A_2 \prec A_3$ | $A_1 \prec A_2 \prec A_3$ |

## 5. Numerical Example

Suppose 4 DMs $(D_h, h = 1, 2, 3, 4)$ of an accelerator must establish criteria and analyze the criteria's effect on a technology-based acceleration program. To achieve this goal, the methods DEMATEL and ANP are performed. Assume $(C_m, m = 1, 2, \ldots, 19)$ are the qualitative criteria and quantitative criteria under consideration, as shown in Figure A1 (see Appendix B for details). Assuming that DMs have reached a consensus, the effects of criteria on each other are indicated using a scale of *No Effect (1)*, *Low Effect (2)*, *Medium Low Effect (3)*, *Medium Effect (4)*, *Medium High Effect (5)*, *High Effect (6)*, and *Extremely Strong Effect (7)*. After each DM rates the alternatives, the aggregating direct-relation matrix is determined using Equation (18) and is shown in Table A1 (see Appendix C for details).

Subsequently, values of the normalized direct-relation matrix are obtained using Equations (19) and (20) and are shown in Table A2 (see Appendix D for details). Finally, the total-relation matrix is attained using Equation (21), as shown in Table A3 (see Appendix E for details). Next, the prominence $(D + R)$ and relation $(D - R)$ values are calculated using Equations (22) and (23). Thereafter, the threshold value is set, which determines the filtered factors. The causal relationship and notable factors are displayed in Table 6 and Figure 2. According to Table 6, "$(C_6)$ demand validation" has the greatest $(D + R)$ value and is the most critical factor, followed by "$(C_7)$ customer affordability" and "$(C_8)$ market demographic". All these factors need to be evaluated in the initial steps when building a product or service. Additionally, the $(D - R)$ values of "$(C_3)$ prior startup experience", "$(C_1)$ sales", and "$(C_2)$ product development cost" demonstrate that these

criteria have net influences on other factors. Other medium value factors that are selected when proceeding to the next steps are "$(C_9)$ concept maturity", "$(C_{10})$ product maturity", "$(C_{11})$ value proposition", "$(C_{13})$ technology experience", "$(C_{15})$ growth strategy", "$(C_{18})$ creativity", and "$(C_{19})$ negotiation".

**Table 6.** Prominence and Relation value of criteria.

|  | *D* | *R* | *D + R* | *D − R* |
|---|---|---|---|---|
| $C_1$ | 1.4660 | 1.0098 | 2.476 | 0.4562 |
| $C_2$ | 1.3166 | 0.9326 | 2.249 | 0.3840 |
| $C_3$ | 2.8245 | 2.0101 | 4.835 | 0.8144 |
| $C_4$ | 1.3291 | 1.9605 | 3.290 | −0.6314 |
| $C_5$ | 1.5740 | 2.1332 | 3.707 | −0.5593 |
| $C_6$ | 3.0201 | 3.1850 | 6.205 | −0.1649 |
| $C_7$ | 2.9359 | 3.1138 | 6.050 | −0.1778 |
| $C_8$ | 2.9104 | 3.1088 | 6.019 | −0.1985 |
| $C_9$ | 2.5804 | 2.8768 | 5.457 | −0.2964 |
| $C_{10}$ | 2.3358 | 2.7069 | 5.043 | −0.3711 |
| $C_{11}$ | 2.2284 | 2.6253 | 4.854 | −0.3969 |
| $C_{12}$ | 1.3718 | 1.9929 | 3.365 | −0.6211 |
| $C_{13}$ | 2.6701 | 2.9201 | 5.590 | −0.2500 |
| $C_{14}$ | 1.3602 | 2.0277 | 3.388 | −0.6675 |
| $C_{15}$ | 2.1349 | 2.5318 | 4.667 | −0.3969 |
| $C_{16}$ | 1.6294 | 2.1784 | 3.808 | −0.5490 |
| $C_{17}$ | 1.8855 | 2.3726 | 4.258 | −0.4871 |
| $C_{18}$ | 2.4492 | 2.7714 | 5.221 | −0.3222 |
| $C_{19}$ | 2.5088 | 2.8181 | 5.327 | −0.3093 |
|  |  | Average | 4.516 |  |

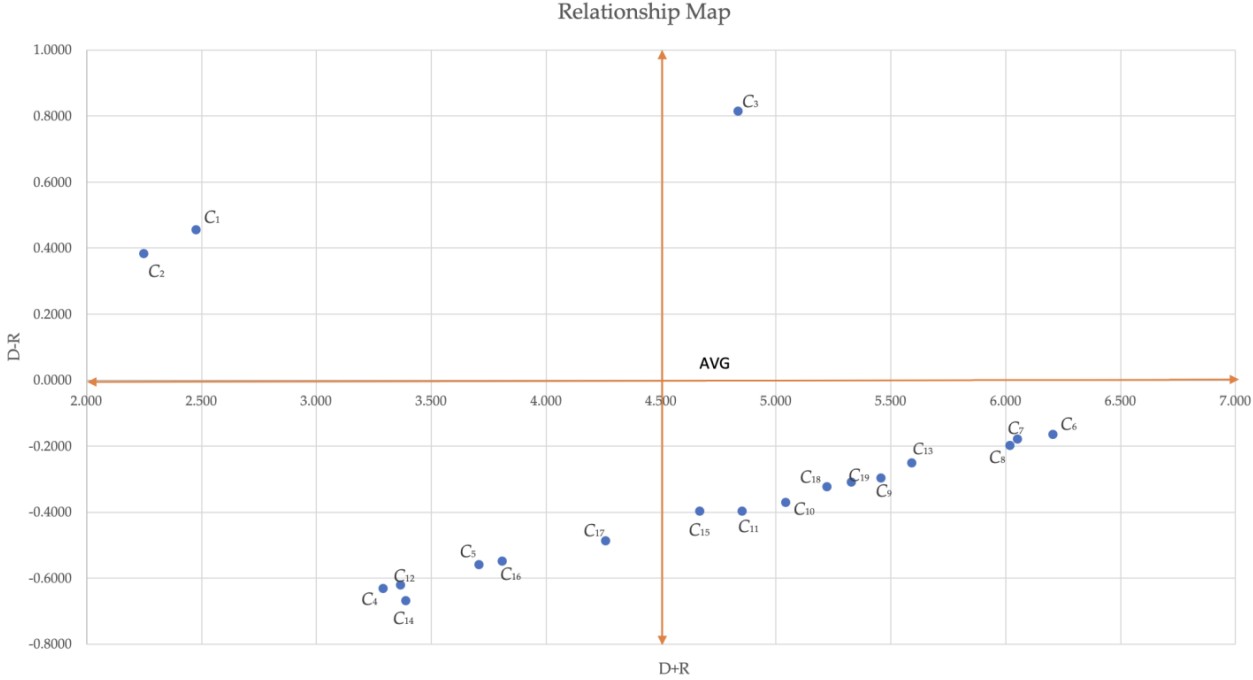

**Figure 2.** Causal Diagram.

Next, the pairwise comparison must be carefully evaluated by DMs according to the criteria. In this study, the statistical software Super Decisions was used for the analysis. Super Decisions is a decision-support program that implements AHP and ANP to calculate the weights of the dimensions and tests the expert's competency. After obtaining the integrated PCM, the values are entered into the software to compute $CR$ values. First, the integrated matrix is computed with respect to each criterion, including the Consistency Ratio $CR \leq 0.1$, as shown in Equations (24)–(28) (see Tables A4–A16 in Appendix F for details). Then, the unweighted supermatrix and weighted matrix are created, as shown in Tables 7 and 8. Finally, the limited matrix with the stable weights and the final weight order can be determined, as shown in Tables 9 and 10. According to Table 10, "($C_8$) market demographics" has the highest value with 0.1253, followed by "($C_6$) demand validation" with 0.1196 and "($C_3$) prior startup experience" with 0.0940. The lowest weight value is "($C_{11}$) value proposition" with 0.0215.

**Table 7.** The unweighted supermatrix.

|  | C1S | C2 PDC | C3 PSE | C6 DV | C7 CA | C8 MD | C9 CM | C10 PM | C11 VP | C13 TE | C15 GS | C18 Cre | C19 Neg |
|---|---|---|---|---|---|---|---|---|---|---|---|---|---|
| C1S | 0.03614 | 0.02318 | 0.17330 | 0.03728 | 0.03882 | 0.03972 | 0.14571 | 0.15206 | 0.03883 | 0.17549 | 0.03695 | 0.04388 | 0.18769 |
| C2 PDC | 0.01653 | 0.04581 | 0.10342 | 0.04958 | 0.04628 | 0.04749 | 0.10732 | 0.10631 | 0.02190 | 0.10314 | 0.01747 | 0.01689 | 0.10480 |
| C3 PSE | 0.03338 | 0.15625 | 0.04742 | 0.17850 | 0.17685 | 0.17666 | 0.03819 | 0.04569 | 0.03989 | 0.04565 | 0.03391 | 0.03338 | 0.04480 |
| C6 DV | 0.01423 | 0.18950 | 0.13997 | 0.15548 | 0.14938 | 0.13554 | 0.14084 | 0.13822 | 0.01601 | 0.15156 | 0.01444 | 0.01552 | 0.13387 |
| C7 CA | 0.16286 | 0.07300 | 0.06280 | 0.08022 | 0.08345 | 0.08509 | 0.06857 | 0.06773 | 0.14256 | 0.04789 | 0.16540 | 0.16458 | 0.04802 |
| C8 MD | 0.12483 | 0.16978 | 0.06602 | 0.18612 | 0.18717 | 0.18934 | 0.06726 | 0.06685 | 0.10806 | 0.06635 | 0.12572 | 0.12298 | 0.06639 |
| C9 CM | 0.03999 | 0.06738 | 0.02107 | 0.10666 | 0.10801 | 0.11045 | 0.02145 | 0.02110 | 0.03401 | 0.02129 | 0.03741 | 0.03359 | 0.02160 |
| C10 PM | 0.07094 | 0.07526 | 0.08248 | 0.08041 | 0.08405 | 0.08596 | 0.08810 | 0.08796 | 0.06341 | 0.08393 | 0.06744 | 0.06902 | 0.08430 |
| C11 VP | 0.02344 | 0.01298 | 0.02558 | 0.01390 | 0.01398 | 0.01415 | 0.02695 | 0.02594 | 0.06117 | 0.02575 | 0.02421 | 0.02427 | 0.02577 |
| C13 TE | 0.22517 | 0.05499 | 0.06987 | 0.03349 | 0.03341 | 0.03477 | 0.08177 | 0.07892 | 0.21790 | 0.07524 | 0.22318 | 0.22273 | 0.07842 |
| C15 GS | 0.06504 | 0.01957 | 0.05668 | 0.01696 | 0.01707 | 0.01769 | 0.05688 | 0.05184 | 0.06853 | 0.05801 | 0.06724 | 0.06790 | 0.05799 |
| C18 Cre | 0.11064 | 0.09415 | 0.02974 | 0.03832 | 0.03826 | 0.03718 | 0.03954 | 0.03792 | 0.10767 | 0.03890 | 0.10966 | 0.10762 | 0.03941 |
| C19 Neg | 0.07680 | 0.01817 | 0.12165 | 0.02307 | 0.02327 | 0.02597 | 0.11742 | 0.11946 | 0.08005 | 0.10681 | 0.07698 | 0.07764 | 0.10694 |

**Table 8.** The weighted supermatrix.

|  | C1 S | C2 PDC | C3 PSE | C6 DV | C7CA | C8 MD | C9 CM | C10 PM | C11 VP | C13 TE | C15 GS | C18 Cre | C19 Neg |
|---|---|---|---|---|---|---|---|---|---|---|---|---|---|
| C1 S | 0.03614 | 0.02318 | 0.17330 | 0.03728 | 0.03882 | 0.03972 | 0.14571 | 0.15206 | 0.03883 | 0.17549 | 0.03695 | 0.04388 | 0.18769 |
| C2 PDC | 0.01653 | 0.04581 | 0.10342 | 0.04958 | 0.04628 | 0.04749 | 0.10732 | 0.10631 | 0.02190 | 0.10314 | 0.01747 | 0.01689 | 0.10480 |
| C3 PSE | 0.03338 | 0.15625 | 0.04742 | 0.17850 | 0.17685 | 0.17666 | 0.03819 | 0.04569 | 0.03989 | 0.04565 | 0.03391 | 0.03338 | 0.04480 |
| C6 DV | 0.01423 | 0.18950 | 0.13997 | 0.15548 | 0.14938 | 0.13554 | 0.14084 | 0.13822 | 0.01601 | 0.15156 | 0.01444 | 0.01552 | 0.13387 |
| C7CA | 0.16286 | 0.07300 | 0.06280 | 0.08022 | 0.08345 | 0.08509 | 0.06857 | 0.06773 | 0.14256 | 0.04789 | 0.16540 | 0.16458 | 0.04802 |
| C8 MD | 0.12483 | 0.16978 | 0.06602 | 0.18612 | 0.18717 | 0.18934 | 0.06726 | 0.06685 | 0.10806 | 0.06635 | 0.12572 | 0.12298 | 0.06639 |
| C9 CM | 0.03999 | 0.06738 | 0.02107 | 0.10666 | 0.10801 | 0.11045 | 0.02145 | 0.02110 | 0.03401 | 0.02129 | 0.03741 | 0.03359 | 0.02160 |
| C10 PM | 0.07094 | 0.07526 | 0.08248 | 0.08041 | 0.08405 | 0.08596 | 0.08810 | 0.08796 | 0.06341 | 0.08393 | 0.06744 | 0.06902 | 0.08430 |
| C11 VP | 0.02344 | 0.01298 | 0.02558 | 0.01390 | 0.01398 | 0.01415 | 0.02695 | 0.02594 | 0.06117 | 0.02575 | 0.02421 | 0.02427 | 0.02577 |
| C13 TE | 0.22517 | 0.05499 | 0.06987 | 0.03349 | 0.03341 | 0.03477 | 0.08177 | 0.07892 | 0.21790 | 0.07524 | 0.22318 | 0.22273 | 0.07842 |
| C15 GS | 0.06504 | 0.01957 | 0.05668 | 0.01696 | 0.01707 | 0.01769 | 0.05688 | 0.05184 | 0.06853 | 0.05801 | 0.06724 | 0.06790 | 0.05799 |
| C18 Cre | 0.11064 | 0.09415 | 0.02974 | 0.03832 | 0.03826 | 0.03718 | 0.03954 | 0.03792 | 0.10767 | 0.03890 | 0.10966 | 0.10762 | 0.03941 |
| C19 Neg | 0.07680 | 0.01817 | 0.12165 | 0.02307 | 0.02327 | 0.02597 | 0.11742 | 0.11946 | 0.08005 | 0.10681 | 0.07698 | 0.07764 | 0.10694 |

**Table 9.** The limited supermatrix.

| | C1 S | C2 PDC | C3 PSE | C6 DV | C7 CA | C8 MD | C9 CM | C10 PM | C11 VP | C13 TE | C15 GS | C18 Cre | C19 Neg |
|---|---|---|---|---|---|---|---|---|---|---|---|---|---|
| C1 S | 0.08852 | 0.08852 | 0.08852 | 0.08852 | 0.08852 | 0.08852 | 0.08852 | 0.08852 | 0.08852 | 0.08852 | 0.08852 | 0.08852 | 0.08852 |
| C2 PDC | 0.06374 | 0.06374 | 0.06374 | 0.06374 | 0.06374 | 0.06374 | 0.06374 | 0.06374 | 0.06374 | 0.06374 | 0.06374 | 0.06374 | 0.06374 |
| C3 PSE | 0.09400 | 0.09400 | 0.09400 | 0.09400 | 0.09400 | 0.09400 | 0.09400 | 0.09400 | 0.09400 | 0.09400 | 0.09400 | 0.09400 | 0.09400 |
| C6 DV | 0.11965 | 0.11965 | 0.11965 | 0.11965 | 0.11965 | 0.11965 | 0.11965 | 0.11965 | 0.11965 | 0.11965 | 0.11965 | 0.11965 | 0.11965 |
| C7 CA | 0.08917 | 0.08917 | 0.08917 | 0.08917 | 0.08917 | 0.08917 | 0.08917 | 0.08917 | 0.08917 | 0.08917 | 0.08917 | 0.08917 | 0.08917 |
| C8 MD | 0.12532 | 0.12532 | 0.12532 | 0.12532 | 0.12532 | 0.12532 | 0.12532 | 0.12532 | 0.12532 | 0.12532 | 0.12532 | 0.12532 | 0.12532 |
| C9 CM | 0.05665 | 0.05665 | 0.05665 | 0.05665 | 0.05665 | 0.05665 | 0.05665 | 0.05665 | 0.05665 | 0.05665 | 0.05665 | 0.05665 | 0.05665 |
| C10 PM | 0.08054 | 0.08054 | 0.08054 | 0.08054 | 0.08054 | 0.08054 | 0.08054 | 0.08054 | 0.08054 | 0.08054 | 0.08054 | 0.08054 | 0.08054 |
| C11 VP | 0.02149 | 0.02149 | 0.02149 | 0.02149 | 0.02149 | 0.02149 | 0.02149 | 0.02149 | 0.02149 | 0.02149 | 0.02149 | 0.02149 | 0.02149 |
| C13 TE | 0.09150 | 0.09150 | 0.09150 | 0.09150 | 0.09150 | 0.09150 | 0.09150 | 0.09150 | 0.09150 | 0.09150 | 0.09150 | 0.09150 | 0.09150 |
| C15 GS | 0.04306 | 0.04306 | 0.04306 | 0.04306 | 0.04306 | 0.04306 | 0.04306 | 0.04306 | 0.04306 | 0.04306 | 0.04306 | 0.04306 | 0.04306 |
| C18 Cre | 0.05593 | 0.05593 | 0.05593 | 0.05593 | 0.05593 | 0.05593 | 0.05593 | 0.05593 | 0.05593 | 0.05593 | 0.05593 | 0.05593 | 0.05593 |
| C19 Neg | 0.07044 | 0.07044 | 0.07044 | 0.07044 | 0.07044 | 0.07044 | 0.07044 | 0.07044 | 0.07044 | 0.07044 | 0.07044 | 0.07044 | 0.07044 |

**Table 10.** Final weight order.

| Criteria | Symbol | Values | Ranking |
|---|---|---|---|
| ($C_8$) Market Demographic | C8 MD | 0.1253 | 1 |
| ($C_6$) Demand Validation | C6 DV | 0.1196 | 2 |
| ($C_3$) Prior Startup Experience | C3 PSE | 0.0940 | 3 |
| ($C_{13}$) Technology Experience | C13 TE | 0.0915 | 4 |
| ($C_7$) Customer affordability | C7 CA | 0.0892 | 5 |
| ($C_1$) Sales | C1 S | 0.0885 | 6 |
| ($C_{10}$) Product Maturity | C10 PM | 0.0805 | 7 |
| ($C_{19}$) Negotiation | C19 Neg | 0.0704 | 8 |
| ($C_2$) Product Development Cost | C2 PDC | 0.0637 | 9 |
| ($C_9$) Concept Maturity | C9 CM | 0.0567 | 10 |
| ($C_{18}$) Creativity | C18 Cre | 0.0559 | 11 |
| ($C_{15}$) Growth Strategy | C15 GS | 0.0431 | 12 |
| ($C_{11}$) Value Proposition | C11 VP | 0.0215 | 13 |

Finally, the fuzzy PROMETHEE II-based spread area ranking method is applied. Suppose the same DM group assesses four technology-based startup projects $(A_n, n = 1, 2, 3, 4)$ under 13 criteria that are screened during the previous steps. The ratings of the alternatives over qualitative criteria and quantitative criteria are shown in Tables A17 and A18 (see Appendices G and H, respectively, for details). Subsequently, the mean ratings are calculated using Equation (29), as shown in Table 11, and the alternatives' normalized gradings versus quantitative criteria are produced using Equations (30) and (31), as shown in Table 12. The confidence level ratings on alternatives are also collected to produce $\mu$ value, as shown in Table 13.

The aggregated fuzzy preference is attained using Equations (32)–(36), as shown in Table 14. Subsequently, the fuzzy leaving flow $\varphi'^{+}(A_i)$, the fuzzy entering flow $\varphi'^{-}(A_i)$, and the fuzzy net outranking flow for each alternative are computed using Equations (37)–(39), as presented in Table 15. Using the proposed spread area-based RMMS model, the fuzzy net outranking flow of each alternative is defuzzified using Equations (9)–(17) and yields values of $A_1$ (−0.0519), $A_2$ (0.0905), $A_3$ (0.0594) and $A_4$ (−0.0980) as presented in Table 16. The final ranking of four startup projects $A_4 < A_1 < A_3 < A_2$ indicates that startup project $A_2$ has the highest comprehensive potential, followed by startup project $A_3$.

**Table 11.** The average ratings of the alternatives over qualitative criteria.

| $C_n$ | Average Rating | | | | | | | | | | | |
|---|---|---|---|---|---|---|---|---|---|---|---|---|
| | $A_1$ | | | $A_2$ | | | $A_3$ | | | $A_4$ | | |
| | $(aj_1, bj_1, cj_1)$ | | | $(aj_2, bj_2, cj_2)$ | | | $(aj_3, bj_3, cj_3)$ | | | $(aj_4, bj_4, cj_4)$ | | |
| $C_6$ | 0.500 | 0.650 | 0.750 | 0.600 | 0.750 | 0.850 | 0.500 | 0.650 | 0.763 | 0.388 | 0.538 | 0.675 |
| $C_7$ | 0.750 | 0.900 | 1.000 | 0.538 | 0.688 | 0.788 | 0.538 | 0.688 | 0.788 | 0.388 | 0.538 | 0.675 |
| $C_8$ | 0.425 | 0.575 | 0.700 | 0.425 | 0.575 | 0.700 | 0.538 | 0.688 | 0.788 | 0.350 | 0.500 | 0.650 |
| $C_9$ | 0.425 | 0.575 | 0.700 | 0.613 | 0.763 | 0.863 | 0.650 | 0.800 | 0.900 | 0.325 | 0.463 | 0.613 |
| $C_{10}$ | 0.500 | 0.650 | 0.763 | 0.563 | 0.713 | 0.813 | 0.500 | 0.650 | 0.750 | 0.388 | 0.538 | 0.675 |
| $C_{11}$ | 0.463 | 0.613 | 0.725 | 0.438 | 0.575 | 0.688 | 0.375 | 0.500 | 0.638 | 0.425 | 0.575 | 0.700 |
| $C_{13}$ | 0.213 | 0.313 | 0.463 | 0.650 | 0.800 | 0.900 | 0.650 | 0.800 | 0.900 | 0.350 | 0.500 | 0.650 |
| $C_{15}$ | 0.213 | 0.313 | 0.463 | 0.650 | 0.800 | 0.900 | 0.650 | 0.800 | 0.900 | 0.350 | 0.500 | 0.650 |
| $C_{18}$ | 0.388 | 0.538 | 0.675 | 0.500 | 0.650 | 0.775 | 0.613 | 0.763 | 0.863 | 0.188 | 0.288 | 0.438 |
| $C_{19}$ | 0.500 | 0.650 | 0.750 | 0.388 | 0.538 | 0.675 | 0.425 | 0.575 | 0.700 | 0.350 | 0.500 | 0.650 |

**Table 12.** The average ratings of the alternatives over quantitative criteria.

| $C_n$ | Average Rating | | | | | | | | | | | |
|---|---|---|---|---|---|---|---|---|---|---|---|---|
| | $A_1$ | | | $A_2$ | | | $A_3$ | | | $A_4$ | | |
| | $(al_1, bl_1, cl_1)$ | | | $(al_2, bl_2, cl_2)$ | | | $(al_3, bl_3, cl_3)$ | | | $(al_4, bl_4, cl_4)$ | | |
| $C_1$ | 0.250 | 0.375 | 0.500 | 0.750 | 0.875 | 1.000 | 0.500 | 0.625 | 0.750 | 0.000 | 0.125 | 0.250 |
| $C_2$ | 0.752 | 0.877 | 1.000 | 0.000 | 0.125 | 0.248 | 0.501 | 0.627 | 0.749 | 0.750 | 0.875 | 1.000 |
| $C_3$ | 0.000 | 0.125 | 0.250 | 0.750 | 0.875 | 1.000 | 0.375 | 0.500 | 0.625 | 0.375 | 0.500 | 0.625 |

**Table 13.** Confidence level $\mu$ from DMs.

| | $A_1$ | $A_2$ | $A_3$ | $A_4$ | | $A_1$ | $A_2$ | $A_3$ | $A_4$ | $\mu$ |
|---|---|---|---|---|---|---|---|---|---|---|
| $D_1$ | 0.6 | 0.8 | 0.7 | 0.6 | $D_3$ | 0.7 | 0.7 | 0.8 | 0.5 | |
| $D_2$ | 0.6 | 0.8 | 0.7 | 0.5 | $D_4$ | 0.5 | 0.8 | 0.7 | 0.6 | 0.6625 |

**Table 14.** The aggregated fuzzy TNs preference.

| | $A_1$ | | | $A_2$ | | | $A_3$ | | | $A_4$ | | |
|---|---|---|---|---|---|---|---|---|---|---|---|---|
| $A_1$ | - | - | - | 0.0321 | 0.0479 | 0.0637 | 0.0002 | 0.0160 | 0.0318 | 0.0164 | 0.0771 | 0.1301 |
| $A_2$ | 0.0863 | 0.1594 | 0.1998 | - | - | - | 0.0118 | 0.0574 | 0.1030 | 0.0628 | 0.1825 | 0.2857 |
| $A_3$ | 0.0289 | 0.1020 | 0.1659 | 0.0161 | 0.0320 | 0.0478 | - | - | - | 0.0373 | 0.1336 | 0.2118 |
| $A_4$ | 0.0117 | 0.0352 | 0.0587 | 0.0321 | 0.0479 | 0.0637 | 0.0002 | 0.0160 | 0.0318 | - | - | - |

**Table 15.** The fuzzy TNs net outranking flow for each alternative.

| | $\phi+$ | | | $\phi-$ | | | $\phi$ | | |
|---|---|---|---|---|---|---|---|---|---|
| $A_1$ | 0.0162 | 0.0470 | 0.0752 | 0.0423 | 0.0989 | 0.1415 | −0.1253 | −0.0519 | 0.0329 |
| $A_2$ | 0.0536 | 0.1331 | 0.1962 | 0.0268 | 0.0426 | 0.0584 | −0.0048 | 0.0905 | 0.1694 |
| $A_3$ | 0.0275 | 0.0892 | 0.1418 | 0.0040 | 0.0298 | 0.0555 | −0.0281 | 0.0594 | 0.1378 |
| $A_4$ | 0.0147 | 0.0330 | 0.0514 | 0.0388 | 0.1310 | 0.2092 | −0.1945 | −0.0980 | 0.0126 |

**Table 16.** Defuzzification and ranking of the alternatives.

|       | $\phi$ |        |        | $S_{L1}$ | $S_{L2}$ | $S_{R1}$ | $S_{R2}$ | $V(A_i)$ | Ranking |
|-------|--------|--------|--------|----------|----------|----------|----------|----------|---------|
| $A_1$ | −0.1253 | −0.0519 | 0.0329 | 0.9364 | 0.1733 | 0.6221 | 0.6364 | −0.0519 | 3 |
| $A_2$ | −0.0048 | 0.0905 | 0.1694 | 1.1217 | 0.2415 | 0.6852 | 0.4956 | 0.0905 | 1 |
| $A_3$ | −0.0281 | 0.0594 | 0.1378 | 1.0830 | 0.2263 | 0.6719 | 0.5262 | 0.0594 | 2 |
| $A_4$ | −0.1945 | −0.0980 | 0.0126 | 0.8599 | 0.1462 | 0.6058 | 0.6724 | −0.0980 | 4 |

The utilization of the DEMATEL-ANP-based fuzzy PROMETHEE II provides a comprehensive procedure for ranking alternatives. The DEMATEL investigated the cause–effect relationships between criteria and filtered out the nonsignificant criteria. Subsequently, ANP helped to determine the criteria weights because it permits criterion dependency. Finally, the final ranking was generated by the fuzzy-based PROMETHEE II method, which includes a proposed ranking model to enhance consistency and discrimination ability. The numerical results demonstrated the feasibility of the hybrid model for various decision-making management applications.

## 6. Conclusions

Language has naturally evolved to reflect human judgment and fuzzy ranking is required to turn assessments into decision-making. An extension on ranking FNs using spread area-based RMMS was proposed to improve the applicability and differentiation of the methods of Wang et al. [54], Nejad and Mashinchi [59], and Chu and Nguyen [63]. The algorithm and equations were derived by implementing a ranking method. Comparative examples demonstrated the strengths of the proposed method in discriminating fuzzy numbers and consistency ranking. Finally, the suggested ranking method was integrated into a hybrid DEMATEL-ANP-based fuzzy PROMETHEE II model to inspect the interrelationships among factors, obtain critical criteria weights, and organize startups for a comprehensive decision-making procedure. The numerical example has illustrated the feasibility of the hybrid fuzzy MCDM method.

In future studies, the proposed fuzzy ranking method can be amalgamated into different MCDM methods to further investigate its validity and apply the method to various practices in entrepreneurial problems, such as project selections, business investment evaluation, accelerator evaluation, risk management, performance evaluation, and other areas where decision-making involves subjective judgment and uncertainty. Hybrid fuzzy ranking methods enable comprehensive evaluation and prioritization of project proposals or initiatives by considering multiple criteria and incorporating fuzzy logic, aiding decision-makers in selecting projects aligned with their strategic objectives. In addition, fuzzy ranking methods can aid in evaluating and comparing different accelerators based on their offerings, mentorship quality, network strength, success rate, and other relevant criteria. This helps entrepreneurs make informed decisions about which accelerator program would best suit their needs and increase their chances of success. The fuzzy ranking approach adds a layer of flexibility to handle uncertain or imprecise data in investment decision-making.

**Author Contributions:** Conceptualization, H.T.N. and T.-C.C.; methodology, H.T.N. and T.-C.C.; validation, H.T.N. and T.-C.C.; formal analysis, H.T.N.; investigation, H.T.N. and T.-C.C.; resources, T.-C.C.; data curation, H.T.N.; writing—original draft preparation, H.T.N.; writing—review and editing, H.T.N. and T.-C.C.; visualization, H.T.N.; supervision, T.-C.C.; project administration, T.-C.C. All authors have read and agreed to the published version of the manuscript.

**Funding:** This work was supported in part by the National Science and Technology Council, Taiwan, under Grant MOST 111-2410-H-218-004.

**Institutional Review Board Statement:** Not applicable.

**Informed Consent Statement:** Not applicable.

**Data Availability Statement:** Not applicable.

**Acknowledgments:** The authors would like to thank the anonymous reviewers for their constructive comments which improved the presentation of this work. This work was supported in part by the National Science and Technology Council, Taiwan, under Grant MOST 111-2410-H-218-004.

**Conflicts of Interest:** The authors declare no conflict of interest.

## Appendix A

The derivation of Equation (A1) for the second left spread area $S_{L_{i2}}$ is presented as follows.

$$
\begin{aligned}
S_{L_{i2}}(A_i) &= \int_{x''_{\min}}^{x_{L_{i2}}} f''_M(x)dx \\
&= \int_{x''_{\min}}^{x_{L_{i2}}} \left( \frac{x - x''_{\min}}{x''_{\max} - x''_{\min}} \right) dx \\
&= \left( \frac{x^2}{2(x''_{\max} - x''_{\min})} - \frac{2xx''_{\min}}{2(x''_{\max} - x''_{\min})} \right) \Bigg|_{x''_{\min}}^{x_{L_{i2}}} = \frac{x_{L_{i2}}^2 - 2x_{L_{i2}}x''_{\min}}{2(x''_{\max} - x''_{\min})} - \frac{(-x''_{\min}{}^2)}{2(x''_{\max} - x''_{\min})} \\
&= \frac{x_{L_{i2}}^2 - 2x_{L_{i2}}x''_{\min} + x''_{\min}{}^2}{2(x''_{\max} - x''_{\min})} = \frac{(x_{L_{i2}} - x''_{\min})^2}{2(x''_{\max} - x''_{\min})}
\end{aligned}
\tag{A1}
$$

The derivation of Equation (A2) for the first right spread area $S_{R_{i1}}$ is presented as follows.

$$
\begin{aligned}
S_{R_{i1}}(A_i) &= \int_{x_{R_{i1}}}^{x''_{\max}} f''_M(x)dx \\
&= \int_{x_{R_{i1}}}^{x''_{\max}} \left( \frac{x - x''_{\min}}{x''_{\max} - x''_{\min}} \right) dx \\
&= \left( \frac{x^2}{2(x''_{\max} - x''_{\min})} - \frac{2xx''_{\min}}{2(x''_{\max} - x''_{\min})} \right) \Bigg|_{x_{R_{i1}}}^{x''_{\max}} = \frac{(x''_{\max}{}^2 - 2x''_{\max}x''_{\min})}{2(x''_{\max} - x''_{\min})} - \frac{x_{R_{i1}}^2 - 2x_{R_{i1}}x''_{\min}}{2(x''_{\max} - x''_{\min})} \\
&= \frac{(x''_{\max} - x_{R_{i1}})(x''_{\max} + x_{R_{i1}}) - 2x''_{\min}(x''_{\max} + x_{R_{i1}})}{2(x''_{\max} - x''_{\min})} = \frac{(x''_{\max} + x_{R_{i1}})(x''_{\max} - x_{R_{i1}} - 2x''_{\min})}{2(x''_{\max} - x''_{\min})}
\end{aligned}
\tag{A2}
$$

The derivation of Equation (A3) for the second right spread area $S_{R_{i2}}$ is presented as follows.

$$
\begin{aligned}
S_{R_{i2}}(A_i) &= \int_{x_{R_{i2}}}^{x''_{\max}} 1\,dx - \int_{x_{R_{i2}}}^{x''_{\max}} f''_N(x)dx \\
&= x''_{\max} - x_{R_{i2}} - \int_{x_{R_{i2}}}^{x''_{\max}} \left( \frac{x - x''_{\max}}{x''_{\min} - x''_{\max}} \right) dx \\
&= x''_{\max} - x_{R_{i2}} - \left( \frac{x^2}{2(x''_{\min} - x''_{\max})} - \frac{xx''_{\max}}{x''_{\min} - x''_{\max}} \right) \Bigg|_{x_{R_{i2}}}^{x''_{\max}} \\
&= x''_{\max} - x_{R_{i2}} - \left( \frac{x''_{\max}{}^2 - 2x''_{\max}{}^2}{2(x''_{\min} - x''_{\max})} - \frac{x_{R_{i2}}^2 - 2x_{R_{i2}}x''_{\max}}{2(x''_{\min} - x''_{\max})} \right) \\
&= \frac{\left(x''_{\max} - x_{R_{i2}}\right)\left(2\left(x''_{\min} - x''_{\max}\right) + \left(x''_{\max} + x_{R_{i2}}\right)\right)}{2\left(x''_{\min} - x''_{\max}\right)} = \frac{\left(x''_{\max} - x_{R_{i2}}\right)\left(2x''_{\min} - x''_{\max} + x_{R_{i2}}\right)}{2\left(x''_{\min} - x''_{\max}\right)}
\end{aligned}
\tag{A3}
$$

## Appendix B

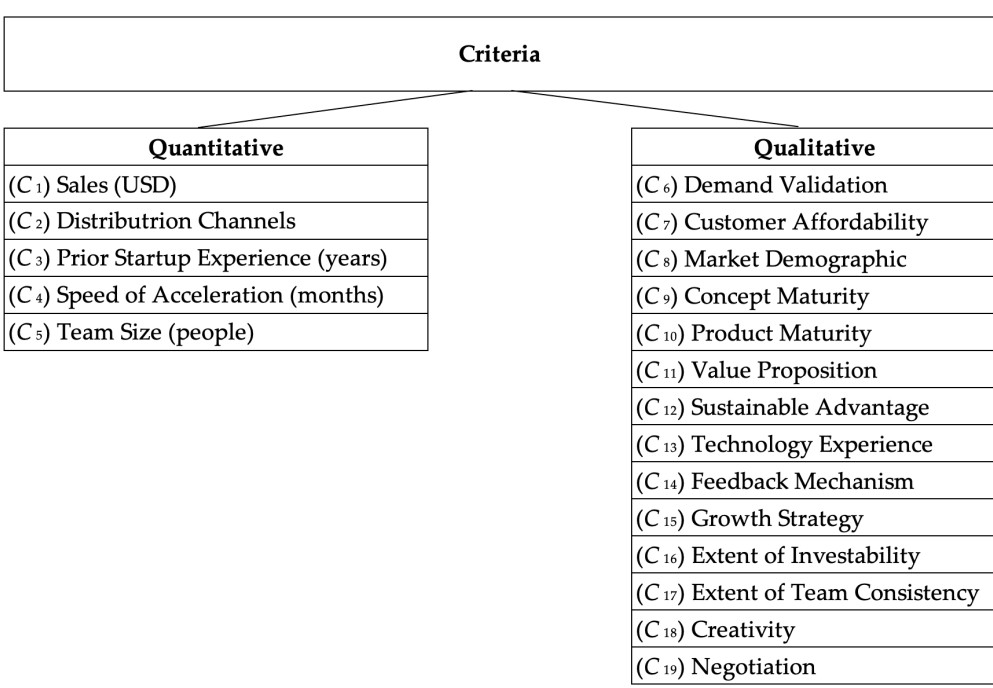

**Figure A1.** Structure of criteria (Yin and Luo, 2018 [5]; Mariño-Garrido et al., 2020 [14]).

## Appendix C

**Table A1.** The aggregating direct-relation matrix of decision-makers.

|  | $C_1$ | $C_2$ | $C_3$ | $C_4$ | $C_5$ | $C_6$ | $C_7$ | $C_8$ | $C_9$ | $C_{10}$ | $C_{11}$ | $C_{12}$ | $C_{13}$ | $C_{14}$ | $C_{15}$ | $C_{16}$ | $C_{17}$ | $C_{18}$ | $C_{19}$ |
|---|---|---|---|---|---|---|---|---|---|---|---|---|---|---|---|---|---|---|---|
| $C_1$ | 0 | 2 | 1.5 | 4 | 5 | 1 | 1 | 1 | 1.5 | 1 | 2.75 | 4 | 4 | 3 | 2 | 5 | 3 | 3 | 3 |
| $C_2$ | 5.5 | 0 | 1.25 | 5 | 4 | 1.25 | 1 | 1.75 | 1.25 | 2 | 1 | 4 | 2 | 3 | 1 | 4 | 2 | 2 | 2 |
| $C_3$ | 6 | 6 | 0 | 6 | 6 | 4 | 5 | 4 | 3 | 3 | 4 | 6 | 5 | 6 | 6 | 5 | 6 | 5 | 5 |
| $C_4$ | 4 | 3 | 1 | 0 | 4 | 1 | 1.75 | 1 | 2 | 1 | 2 | 3 | 1 | 5 | 2 | 3 | 3 | 4 | 2 |
| $C_5$ | 3 | 4 | 1 | 4 | 0 | 1.75 | 2 | 2 | 1 | 2 | 1 | 4 | 4 | 6 | 1 | 2 | 3 | 5 | 4 |
| $C_6$ | 6 | 6 | 4 | 5.75 | 6 | 0 | 4.25 | 4.5 | 6 | 6 | 6 | 6 | 4 | 5.75 | 6 | 6 | 5.75 | 5.25 | 3.75 |
| $C_7$ | 6 | 6 | 3 | 6 | 6 | 3.75 | 0 | 3.25 | 6 | 5.75 | 5.75 | 6 | 6 | 6 | 5.25 | 5.5 | 6 | 4.75 | 3.75 |
| $C_8$ | 5.75 | 6 | 4 | 5.75 | 5.25 | 3.5 | 4.75 | 0 | 5.5 | 5.25 | 6 | 5.5 | 4 | 6 | 5.75 | 6 | 5.75 | 5 | 3.75 |
| $C_9$ | 6 | 6 | 5 | 6 | 6 | 1.5 | 1 | 2.5 | 0 | 6 | 5 | 6 | 3.25 | 6 | 6 | 6 | 5 | 4 | 4 |
| $C_{10}$ | 6 | 6 | 5 | 6 | 5.75 | 1 | 1 | 2.25 | 2 | 0 | 5 | 6 | 3 | 6 | 5 | 5 | 6 | 3 | 4 |
| $C_{11}$ | 5.25 | 6 | 4 | 6 | 6 | 2 | 2 | 1.75 | 2.75 | 3 | 0 | 6 | 5 | 5 | 3 | 6 | 4 | 3 | 3 |
| $C_{12}$ | 4 | 4 | 1 | 4.75 | 4 | 1 | 1 | 1 | 1 | 2 | 2 | 0 | 2 | 3 | 2 | 4 | 4 | 2 | 3 |
| $C_{13}$ | 4 | 6 | 3 | 6 | 4 | 4 | 2 | 4 | 4.75 | 5 | 3 | 6 | 0 | 6 | 6 | 5 | 5 | 6 | 6 |
| $C_{14}$ | 5 | 5 | 2 | 3 | 2 | 2.25 | 2 | 1.5 | 1.75 | 1.75 | 3 | 5 | 1 | 0 | 1 | 2 | 2 | 1 | 3 |
| $C_{15}$ | 6 | 6 | 2 | 6 | 6 | 1 | 2.75 | 2 | 1.75 | 3 | 5 | 6 | 2 | 6 | 0 | 6 | 6 | 2 | 3 |
| $C_{16}$ | 3 | 4 | 3 | 5 | 6 | 2 | 1.75 | 1 | 2 | 3 | 2 | 4 | 3 | 6 | 1 | 0 | 3 | 2 | 2 |
| $C_{17}$ | 5 | 6 | 2 | 5 | 5 | 1.75 | 2 | 2 | 3 | 2 | 4 | 4 | 3 | 6 | 2 | 5 | 0 | 3 | 2 |
| $C_{18}$ | 4.75 | 5.75 | 3 | 4 | 3 | 2.75 | 3 | 3 | 4 | 5 | 5 | 6 | 2 | 6 | 6 | 6 | 5 | 0 | 5 |
| $C_{19}$ | 5 | 6 | 3 | 6 | 4 | 4 | 4 | 4 | 4 | 4 | 5 | 5 | 2 | 5 | 5 | 6 | 6 | 3 | 0 |

## Appendix D

**Table A2.** The normalized direct-relation matrix.

| | $C_1$ | $C_2$ | $C_3$ | $C_4$ | $C_5$ | $C_6$ | $C_7$ | $C_8$ | $C_9$ | $C_{10}$ | $C_{11}$ | $C_{12}$ | $C_{13}$ | $C_{14}$ | $C_{15}$ | $C_{16}$ | $C_{17}$ | $C_{18}$ | $C_{19}$ |
|---|---|---|---|---|---|---|---|---|---|---|---|---|---|---|---|---|---|---|---|
| $C_1$ | 0 | 0.0206 | 0.0155 | 0.0412 | 0.0515 | 0.0103 | 0.0103 | 0.0103 | 0.0155 | 0.0103 | 0.0284 | 0.0412 | 0.0412 | 0.0309 | 0.0206 | 0.0515 | 0.0309 | 0.0309 | 0.0309 |
| $C_2$ | 0.0567 | 0 | 0.0129 | 0.0515 | 0.0412 | 0.0129 | 0.0103 | 0.0180 | 0.0129 | 0.0206 | 0.0103 | 0.0412 | 0.0206 | 0.0309 | 0.0103 | 0.0412 | 0.0206 | 0.0206 | 0.0206 |
| $C_3$ | 0.0619 | 0.0619 | 0 | 0.0619 | 0.0619 | 0.0412 | 0.0515 | 0.0412 | 0.0309 | 0.0309 | 0.0412 | 0.0619 | 0.0515 | 0.0619 | 0.0619 | 0.0515 | 0.0619 | 0.0515 | 0.0515 |
| $C_4$ | 0.0412 | 0.0309 | 0.0103 | 0 | 0.0412 | 0.0103 | 0.0180 | 0.0103 | 0.0206 | 0.0103 | 0.0206 | 0.0309 | 0.0103 | 0.0515 | 0.0206 | 0.0309 | 0.0309 | 0.0412 | 0.0206 |
| $C_5$ | 0.0309 | 0.0412 | 0.0103 | 0.0412 | 0 | 0.0180 | 0.0206 | 0.0206 | 0.0103 | 0.0206 | 0.0103 | 0.0412 | 0.0412 | 0.0619 | 0.0103 | 0.0206 | 0.0309 | 0.0515 | 0.0412 |
| $C_6$ | 0.0619 | 0.0619 | 0.0412 | 0.0593 | 0.0619 | 0 | 0.0438 | 0.0464 | 0.0619 | 0.0619 | 0.0619 | 0.0619 | 0.0412 | 0.0593 | 0.0619 | 0.0619 | 0.0593 | 0.0541 | 0.0387 |
| $C_7$ | 0.0619 | 0.0619 | 0.0309 | 0.0619 | 0.0619 | 0.0387 | 0 | 0.0335 | 0.0619 | 0.0593 | 0.0593 | 0.0619 | 0.0619 | 0.0619 | 0.0619 | 0.0541 | 0.0567 | 0.0619 | 0.0490 | 0.0387 |
| $C_8$ | 0.0593 | 0.0619 | 0.0412 | 0.0593 | 0.0541 | 0.0361 | 0.0490 | 0 | 0.0567 | 0.0541 | 0.0619 | 0.0567 | 0.0412 | 0.0619 | 0.0593 | 0.0619 | 0.0593 | 0.0515 | 0.0387 |
| $C_9$ | 0.0619 | 0.0619 | 0.0515 | 0.0619 | 0.0619 | 0.0155 | 0.0103 | 0.0258 | 0 | 0.0619 | 0.0515 | 0.0619 | 0.0335 | 0.0619 | 0.0619 | 0.0619 | 0.0515 | 0.0412 | 0.0412 |
| $C_{10}$ | 0.0619 | 0.0619 | 0.0515 | 0.0619 | 0.0593 | 0.0103 | 0.0103 | 0.0232 | 0.0206 | 0 | 0.0515 | 0.0619 | 0.0309 | 0.0619 | 0.0515 | 0.0515 | 0.0619 | 0.0309 | 0.0412 |
| $C_{11}$ | 0.0541 | 0.0619 | 0.0412 | 0.0619 | 0.0619 | 0.0206 | 0.0206 | 0.0180 | 0.0284 | 0.0309 | 0 | 0.0619 | 0.0515 | 0.0515 | 0.0309 | 0.0619 | 0.0412 | 0.0309 | 0.0309 |
| $C_{12}$ | 0.0412 | 0.0412 | 0.0103 | 0.0490 | 0.0412 | 0.0103 | 0.0103 | 0.0103 | 0.0103 | 0.0206 | 0.0206 | 0 | 0.0206 | 0.0309 | 0.0206 | 0.0412 | 0.0412 | 0.0206 | 0.0309 |
| $C_{13}$ | 0.0412 | 0.0619 | 0.0309 | 0.0619 | 0.0412 | 0.0412 | 0.0206 | 0.0412 | 0.0490 | 0.0515 | 0.0309 | 0.0619 | 0 | 0.0619 | 0.0619 | 0.0515 | 0.0515 | 0.0619 | 0.0619 |
| $C_{14}$ | 0.0515 | 0.0515 | 0.0206 | 0.0309 | 0.0206 | 0.0232 | 0.0206 | 0.0155 | 0.0180 | 0.0180 | 0.0309 | 0.0515 | 0.0103 | 0 | 0.0103 | 0.0206 | 0.0206 | 0.0103 | 0.0309 |
| $C_{15}$ | 0.0619 | 0.0619 | 0.0206 | 0.0619 | 0.0619 | 0.0103 | 0.0284 | 0.0206 | 0.0180 | 0.0309 | 0.0515 | 0.0619 | 0.0206 | 0.0619 | 0 | 0.0619 | 0.0619 | 0.0206 | 0.0309 |
| $C_{16}$ | 0.0309 | 0.0412 | 0.0309 | 0.0515 | 0.0619 | 0.0206 | 0.0180 | 0.0103 | 0.0206 | 0.0309 | 0.0206 | 0.0412 | 0.0309 | 0.0619 | 0.0103 | 0 | 0.0309 | 0.0206 | 0.0206 |
| $C_{17}$ | 0.0515 | 0.0619 | 0.0206 | 0.0515 | 0.0515 | 0.0180 | 0.0206 | 0.0206 | 0.0309 | 0.0206 | 0.0412 | 0.0412 | 0.0309 | 0.0619 | 0.0206 | 0.0515 | 0 | 0.0309 | 0.0206 |
| $C_{18}$ | 0.0490 | 0.0593 | 0.0309 | 0.0412 | 0.0309 | 0.0284 | 0.0309 | 0.0309 | 0.0412 | 0.0515 | 0.0515 | 0.0619 | 0.0206 | 0.0619 | 0.0619 | 0.0619 | 0.0515 | 0 | 0.0515 |
| $C_{19}$ | 0.0515 | 0.0619 | 0.0309 | 0.0619 | 0.0412 | 0.0412 | 0.0412 | 0.0412 | 0.0412 | 0.0412 | 0.0515 | 0.0515 | 0.0206 | 0.0515 | 0.0515 | 0.0619 | 0.0619 | 0.0309 | 0 |

## Appendix E

**Table A3.** The total-relation matrix.

|  | $C_1$ | $C_2$ | $C_3$ | $C_4$ | $C_5$ | $C_6$ | $C_7$ | $C_8$ | $C_9$ | $C_{10}$ | $C_{11}$ | $C_{12}$ | $C_{13}$ | $C_{14}$ | $C_{15}$ | $C_{16}$ | $C_{17}$ | $C_{18}$ | $C_{19}$ |
|---|---|---|---|---|---|---|---|---|---|---|---|---|---|---|---|---|---|---|---|
| $C_1$ | 0.0681 | 0.0911 | 0.0507 | 0.1117 | 0.1165 | 0.0406 | 0.0418 | 0.0417 | 0.0532 | 0.0539 | 0.0754 | 0.1097 | 0.0820 | 0.1040 | 0.0655 | 0.1147 | 0.0891 | 0.0780 | 0.0781 |
| $C_2$ | 0.1152 | 0.0620 | 0.0442 | 0.1134 | 0.1000 | 0.0391 | 0.0381 | 0.0450 | 0.0462 | 0.0580 | 0.0532 | 0.1018 | 0.0581 | 0.0954 | 0.0504 | 0.0979 | 0.0727 | 0.0631 | 0.0629 |
| $C_3$ | 0.1930 | 0.1963 | 0.0692 | 0.1986 | 0.1895 | 0.0979 | 0.1110 | 0.1010 | 0.1050 | 0.1151 | 0.1353 | 0.1951 | 0.1320 | 0.2009 | 0.1486 | 0.1774 | 0.1747 | 0.1419 | 0.1419 |
| $C_4$ | 0.1022 | 0.0939 | 0.0424 | 0.0646 | 0.1001 | 0.0372 | 0.0459 | 0.0383 | 0.0542 | 0.0495 | 0.0642 | 0.0936 | 0.0488 | 0.1156 | 0.0607 | 0.0893 | 0.0830 | 0.0821 | 0.0635 |
| $C_5$ | 0.1045 | 0.1161 | 0.0487 | 0.1166 | 0.0708 | 0.0507 | 0.0543 | 0.0545 | 0.0524 | 0.0675 | 0.0635 | 0.1154 | 0.0841 | 0.1374 | 0.0606 | 0.0910 | 0.0938 | 0.1004 | 0.0917 |
| $C_6$ | 0.2028 | 0.2062 | 0.1154 | 0.2064 | 0.1997 | 0.0614 | 0.1072 | 0.1095 | 0.1383 | 0.1502 | 0.1615 | 0.2051 | 0.1286 | 0.2088 | 0.1554 | 0.1962 | 0.1806 | 0.1503 | 0.1363 |
| $C_7$ | 0.1983 | 0.2019 | 0.1034 | 0.2044 | 0.1952 | 0.0969 | 0.0626 | 0.0957 | 0.1360 | 0.1451 | 0.1556 | 0.2009 | 0.1449 | 0.2067 | 0.1453 | 0.1873 | 0.1790 | 0.1430 | 0.1338 |
| $C_8$ | 0.1952 | 0.2009 | 0.1124 | 0.2009 | 0.1873 | 0.0941 | 0.1096 | 0.0627 | 0.1307 | 0.1396 | 0.1577 | 0.1950 | 0.1254 | 0.2056 | 0.1493 | 0.1912 | 0.1759 | 0.1442 | 0.1326 |
| $C_9$ | 0.1810 | 0.1836 | 0.1125 | 0.1862 | 0.1783 | 0.0671 | 0.0657 | 0.0797 | 0.0652 | 0.1341 | 0.1349 | 0.1828 | 0.1071 | 0.1881 | 0.1389 | 0.1748 | 0.1541 | 0.1226 | 0.1237 |
| $C_{10}$ | 0.1690 | 0.1713 | 0.1054 | 0.1737 | 0.1641 | 0.0575 | 0.0606 | 0.0718 | 0.0791 | 0.0670 | 0.1258 | 0.1703 | 0.0975 | 0.1753 | 0.1205 | 0.1537 | 0.1528 | 0.1050 | 0.1152 |
| $C_{11}$ | 0.1562 | 0.1659 | 0.0934 | 0.1685 | 0.1614 | 0.0655 | 0.0677 | 0.0652 | 0.0846 | 0.0953 | 0.0727 | 0.1653 | 0.1140 | 0.1607 | 0.0988 | 0.1582 | 0.1293 | 0.1028 | 0.1029 |
| $C_{12}$ | 0.1037 | 0.1051 | 0.0431 | 0.1140 | 0.1026 | 0.0378 | 0.0393 | 0.0391 | 0.0452 | 0.0596 | 0.0648 | 0.0649 | 0.0595 | 0.0985 | 0.0614 | 0.1006 | 0.0942 | 0.0645 | 0.0739 |
| $C_{13}$ | 0.1672 | 0.1894 | 0.0965 | 0.1911 | 0.1631 | 0.0941 | 0.0785 | 0.0975 | 0.1172 | 0.1301 | 0.1213 | 0.1880 | 0.0765 | 0.1933 | 0.1446 | 0.1709 | 0.1593 | 0.1453 | 0.1463 |
| $C_{14}$ | 0.1139 | 0.1143 | 0.0532 | 0.0973 | 0.0834 | 0.0500 | 0.0491 | 0.0441 | 0.0528 | 0.0576 | 0.0749 | 0.1143 | 0.0505 | 0.0667 | 0.0526 | 0.0819 | 0.0752 | 0.0545 | 0.0738 |
| $C_{15}$ | 0.1589 | 0.1607 | 0.0711 | 0.1633 | 0.1571 | 0.0529 | 0.0721 | 0.0642 | 0.0714 | 0.0909 | 0.1184 | 0.1600 | 0.0822 | 0.1648 | 0.0632 | 0.1535 | 0.1435 | 0.0885 | 0.0981 |
| $C_{16}$ | 0.1067 | 0.1181 | 0.0692 | 0.1289 | 0.1331 | 0.0533 | 0.0525 | 0.0452 | 0.0619 | 0.0774 | 0.0734 | 0.1175 | 0.0770 | 0.1404 | 0.0606 | 0.0716 | 0.0952 | 0.0737 | 0.0738 |
| $C_{17}$ | 0.1379 | 0.1492 | 0.0659 | 0.1415 | 0.1357 | 0.0558 | 0.0599 | 0.0598 | 0.0781 | 0.0755 | 0.1011 | 0.1297 | 0.0846 | 0.1527 | 0.0777 | 0.1331 | 0.0749 | 0.0909 | 0.0815 |
| $C_{18}$ | 0.1638 | 0.1759 | 0.0910 | 0.1614 | 0.1443 | 0.0770 | 0.0827 | 0.0822 | 0.1032 | 0.1224 | 0.1324 | 0.1772 | 0.0915 | 0.1816 | 0.1359 | 0.1703 | 0.1496 | 0.0782 | 0.1285 |
| $C_{19}$ | 0.1689 | 0.1808 | 0.0923 | 0.1833 | 0.1570 | 0.0904 | 0.0939 | 0.0932 | 0.1056 | 0.1148 | 0.1343 | 0.1701 | 0.0941 | 0.1755 | 0.1281 | 0.1727 | 0.1613 | 0.1115 | 0.0810 |

## Appendix F

**Table A4.** Comparison Matrix of 13 criteria with respect to criterion 1.

|          | $C_1$ | $C_2$ | $C_3$ | $C_6$ | $C_7$ | $C_8$ | $C_9$ | $C_{10}$ | $C_{11}$ | $C_{13}$ | $C_{15}$ | $C_{18}$ | $C_{19}$ |
|----------|-------|-------|-------|-------|-------|-------|-------|----------|----------|----------|----------|----------|----------|
| $C_1$    | 1     | 2     | 3     | 3     | 1/4   | 1/5   | 1/4   | 1/5      | 3        | 1/7      | 1/4      | 1/3      | 1/2      |
| $C_2$    | 1/2   | 1     | 1/3   | 3     | 1/5   | 1/8   | 1/4   | 1/5      | 1/4      | 1/8      | 1/3      | 1/8      | 1/5      |
| $C_3$    | 1/3   | 3     | 1     | 3     | 1/5   | 1/4   | 2     | 1/2      | 3        | 1/6      | 1/3      | 1/3      | 1/6      |
| $C_6$    | 1/3   | 1/3   | 1/3   | 1     | 1/5   | 1/8   | 1/2   | 1/5      | 1/3      | 1/8      | 1/6      | 1/8      | 1/3      |
| $C_7$    | 4     | 5     | 5     | 5     | 1     | 3     | 6     | 3        | 6        | 1/2      | 4        | 2        | 3        |
| $C_8$    | 5     | 8     | 4     | 8     | 1/3   | 1     | 5     | 2        | 5        | 1/3      | 2        | 2        | 3        |
| $C_9$    | 4     | 4     | 1/2   | 2     | 1/6   | 1/5   | 1     | 1/2      | 2        | 1/8      | 1/2      | 1/7      | 1/5      |
| $C_{10}$ | 5     | 5     | 2     | 5     | 1/3   | 1/2   | 2     | 1        | 3        | 1/6      | 1/2      | 1/2      | 2        |
| $C_{11}$ | 1/3   | 4     | 1/3   | 3     | 1/6   | 1/5   | 1/2   | 1/3      | 1        | 1/8      | 1/2      | 1/5      | 1/4      |
| $C_{13}$ | 7     | 8     | 6     | 8     | 2     | 3     | 8     | 6        | 8        | 1        | 4        | 2        | 4        |
| $C_{15}$ | 4     | 3     | 3     | 6     | 1/4   | 1/2   | 2     | 2        | 2        | 1/4      | 1        | 1/2      | 1/2      |
| $C_{18}$ | 3     | 8     | 3     | 8     | 1/2   | 1/2   | 7     | 5        | 5        | 1/2      | 2        | 1        | 2        |
| $C_{19}$ | 2     | 5     | 6     | 3     | 1/3   | 1/3   | 5     | 4        | 4        | 1/4      | 2        | 1/2      | 1        |

Inconsistency: 0.08328

**Table A5.** Comparison Matrix of 13 criteria with respect to criterion 2.

|          | $C_1$ | $C_2$ | $C_3$ | $C_6$ | $C_7$ | $C_8$ | $C_9$ | $C_{10}$ | $C_{11}$ | $C_{13}$ | $C_{15}$ | $C_{18}$ | $C_{19}$ |
|----------|-------|-------|-------|-------|-------|-------|-------|----------|----------|----------|----------|----------|----------|
| $C_1$    | 1     | 1/6   | 1/5   | 1/7   | 1/8   | 1/8   | 1/6   | 1/8      | 3        | 1/2      | 3        | 1/4      | 3        |
| $C_2$    | 6     | 1     | 1/3   | 1/5   | 1/2   | 1/4   | 1/3   | 1/2      | 3        | 1/2      | 4        | 1/2      | 4        |
| $C_3$    | 5     | 3     | 1     | 3     | 3     | 1/2   | 2     | 4        | 7        | 2        | 5        | 2        | 6        |
| $C_6$    | 7     | 5     | 1/3   | 1     | 5     | 3     | 2     | 3        | 9        | 5        | 7        | 3        | 7        |
| $C_7$    | 8     | 2     | 1/3   | 1/5   | 1     | 1/6   | 3     | 1/2      | 5        | 2        | 4        | 1/2      | 4        |
| $C_8$    | 8     | 4     | 2     | 1/3   | 6     | 1     | 3     | 2        | 8        | 6        | 4        | 2        | 6        |
| $C_9$    | 6     | 3     | 1/2   | 1/2   | 1/3   | 1/3   | 1     | 1/2      | 4        | 2        | 4        | 1/3      | 5        |
| $C_{10}$ | 8     | 2     | 1/4   | 1/3   | 2     | 1/2   | 2     | 1        | 6        | 1/3      | 4        | 1/2      | 4        |
| $C_{11}$ | 1/3   | 1/3   | 1/7   | 1/9   | 1/5   | 1/8   | 1/4   | 1/6      | 1        | 1/3      | 1/2      | 1/7      | 1/3      |
| $C_{13}$ | 2     | 2     | 1/2   | 1/5   | 1/2   | 1/6   | 1/2   | 3        | 3        | 1        | 3        | 1/2      | 2        |
| $C_{15}$ | 1/3   | 1/4   | 1/5   | 1/7   | 1/4   | 1/4   | 1/4   | 1/4      | 2        | 1/3      | 1        | 1/6      | 2        |
| $C_{18}$ | 4     | 2     | 1/2   | 1/3   | 2     | 1/2   | 3     | 2        | 7        | 2        | 6        | 1        | 4        |
| $C_{19}$ | 1/3   | 1/4   | 1/6   | 1/7   | 1/4   | 1/6   | 1/5   | 1/4      | 3        | 1/2      | 1/2      | 1/4      | 1        |

Inconsistency: 0.09659

**Table A6.** Comparison Matrix of 13 criteria with respect to criterion 3.

|          | $C_1$ | $C_2$ | $C_3$ | $C_6$ | $C_7$ | $C_8$ | $C_9$ | $C_{10}$ | $C_{11}$ | $C_{13}$ | $C_{15}$ | $C_{18}$ | $C_{19}$ |
|----------|-------|-------|-------|-------|-------|-------|-------|----------|----------|----------|----------|----------|----------|
| $C_1$    | 1     | 3     | 3     | 1/2   | 3     | 2     | 8     | 5        | 6        | 3        | 2        | 4        | 3        |
| $C_2$    | 1/3   | 1     | 3     | 1/2   | 2     | 2     | 4     | 3        | 3        | 2        | 4        | 2        | 1/2      |
| $C_3$    | 1/3   | 1/3   | 1     | 1/3   | 3     | 1/2   | 2     | 1/3      | 2        | 1/3      | 1/2      | 3        | 1/4      |
| $C_6$    | 2     | 2     | 3     | 1     | 1/2   | 2     | 4     | 2        | 4        | 3        | 2        | 5        | 2        |
| $C_7$    | 1/3   | 1/2   | 1/3   | 2     | 1     | 1/2   | 2     | 1/3      | 2        | 1/2      | 3        | 2        | 1/3      |
| $C_8$    | 1/2   | 1/2   | 2     | 1/2   | 2     | 1     | 3     | 1/2      | 3        | 1/2      | 2        | 3        | 1/2      |
| $C_9$    | 1/8   | 1/4   | 1/2   | 1/4   | 1/2   | 1/3   | 1     | 1/4      | 1/2      | 1/3      | 1/2      | 1/2      | 1/4      |
| $C_{10}$ | 1/5   | 1/3   | 3     | 1/2   | 3     | 2     | 4     | 1        | 2        | 3        | 1/2      | 4        | 1/3      |
| $C_{11}$ | 1/6   | 1/3   | 1/2   | 1/4   | 1/2   | 1/3   | 2     | 1/2      | 1        | 1/2      | 1/3      | 1/2      | 1/5      |
| $C_{13}$ | 1/3   | 1/2   | 3     | 1/3   | 2     | 2     | 3     | 1/3      | 2        | 1        | 3        | 1/2      | 2        |
| $C_{15}$ | 1/2   | 1/4   | 2     | 1/2   | 1/3   | 1/2   | 2     | 2        | 3        | 1/3      | 1        | 3        | 1/3      |
| $C_{18}$ | 1/4   | 1/2   | 1/3   | 1/5   | 1/2   | 1/3   | 2     | 1/4      | 2        | 2        | 1/3      | 1        | 1/2      |
| $C_{19}$ | 1/3   | 2     | 4     | 2     | 3     | 2     | 4     | 3        | 5        | 1/2      | 3        | 2        | 1        |
| Inconsistency: 0.09391 | | | | | | | | | | | | | |

**Table A7.** Comparison Matrix of 13 criteria with respect to criterion 6.

|          | $C_1$ | $C_2$ | $C_3$ | $C_6$ | $C_7$ | $C_8$ | $C_9$ | $C_{10}$ | $C_{11}$ | $C_{13}$ | $C_{15}$ | $C_{18}$ | $C_{19}$ |
|----------|-------|-------|-------|-------|-------|-------|-------|----------|----------|----------|----------|----------|----------|
| $C_1$    | 1     | 1/6   | 1/5   | 1/7   | 1/6   | 1/7   | 1/4   | 1/5      | 3        | 2        | 3        | 3        | 4        |
| $C_2$    | 6     | 1     | 1/3   | 1/4   | 1/2   | 1/4   | 1/3   | 1/3      | 3        | 1/2      | 4        | 1/2      | 3        |
| $C_3$    | 5     | 3     | 1     | 4     | 3     | 1/2   | 2     | 4        | 6        | 3        | 4        | 6        | 7        |
| $C_6$    | 7     | 4     | 1/4   | 1     | 4     | 1/2   | 2     | 2        | 7        | 8        | 6        | 6        | 7        |
| $C_7$    | 6     | 2     | 1/3   | 1/4   | 1     | 1/2   | 1/3   | 2        | 5        | 3        | 4        | 3        | 2        |
| $C_8$    | 7     | 4     | 2     | 2     | 2     | 1     | 3     | 2        | 8        | 5        | 8        | 6        | 7        |
| $C_9$    | 4     | 3     | 1/2   | 1/2   | 3     | 1/3   | 1     | 2        | 7        | 3        | 7        | 3        | 5        |
| $C_{10}$ | 5     | 3     | 1/4   | 1/2   | 1/2   | 1/2   | 1/2   | 1        | 5        | 3        | 4        | 3        | 5        |
| $C_{11}$ | 1/3   | 1/3   | 1/6   | 1/7   | 1/5   | 1/8   | 1/7   | 1/5      | 1        | 1/3      | 1/2      | 1/4      | 1/2      |
| $C_{13}$ | 1/2   | 2     | 1/3   | 1/8   | 1/3   | 1/5   | 1/3   | 3        | 3        | 1        | 4        | 1/2      | 1/2      |
| $C_{15}$ | 1/3   | 1/4   | 1/4   | 1/6   | 1/4   | 1/8   | 1/7   | 1/4      | 2        | 1/4      | 1        | 1/3      | 1/2      |
| $C_{18}$ | 1/3   | 2     | 1/6   | 1/6   | 1/3   | 1/6   | 1/3   | 1/3      | 4        | 2        | 3        | 1        | 3        |
| $C_{19}$ | 1/4   | 1/3   | 1/7   | 1/7   | 1/2   | 1/7   | 1/5   | 1/5      | 2        | 2        | 2        | 1/3      | 1        |
| Inconsistency: 0.09784 | | | | | | | | | | | | | |

**Table A8.** Comparison Matrix of 13 criteria with respect to criterion 7.

|  | $C_1$ | $C_2$ | $C_3$ | $C_6$ | $C_7$ | $C_8$ | $C_9$ | $C_{10}$ | $C_{11}$ | $C_{13}$ | $C_{15}$ | $C_{18}$ | $C_{19}$ |
|---|---|---|---|---|---|---|---|---|---|---|---|---|---|
| $C_1$ | 1 | 1/5 | 1/5 | 1/4 | 1/6 | 1/7 | 1/4 | 1/5 | 3 | 2 | 3 | 3 | 4 |
| $C_2$ | 5 | 1 | 1/4 | 1/4 | 1/2 | 1/4 | 1/3 | 1/4 | 3 | 1/2 | 4 | 1/2 | 3 |
| $C_3$ | 5 | 4 | 1 | 4 | 2 | 1/2 | 2 | 4 | 6 | 3 | 4 | 6 | 7 |
| $C_6$ | 4 | 4 | 1/4 | 1 | 4 | 1/2 | 2 | 2 | 7 | 8 | 6 | 6 | 7 |
| $C_7$ | 6 | 2 | 1/2 | 1/4 | 1 | 1/2 | 1/3 | 2 | 5 | 3 | 4 | 3 | 2 |
| $C_8$ | 7 | 4 | 2 | 2 | 2 | 1 | 3 | 2 | 8 | 5 | 8 | 6 | 7 |
| $C_9$ | 4 | 3 | 1/2 | 1/2 | 3 | 1/3 | 1 | 2 | 7 | 3 | 7 | 3 | 5 |
| $C_{10}$ | 5 | 4 | 1/4 | 1/2 | 1/2 | 1/2 | 1/2 | 1 | 5 | 3 | 4 | 3 | 5 |
| $C_{11}$ | 1/3 | 1/3 | 1/6 | 1/7 | 1/5 | 1/8 | 1/7 | 1/5 | 1 | 1/3 | 1/2 | 1/4 | 1/2 |
| $C_{13}$ | 1/2 | 2 | 1/3 | 1/8 | 1/3 | 1/5 | 1/3 | 3 | 3 | 1 | 4 | 1/2 | 1/2 |
| $C_{15}$ | 1/3 | 1/4 | 1/4 | 1/6 | 1/4 | 1/8 | 1/7 | 1/4 | 2 | 1/4 | 1 | 1/3 | 1/2 |
| $C_{18}$ | 1/3 | 2 | 1/6 | 1/6 | 1/3 | 1/6 | 1/3 | 1/3 | 4 | 2 | 3 | 1 | 3 |
| $C_{19}$ | 1/4 | 1/3 | 1/7 | 1/7 | 1/2 | 1/7 | 1/5 | 1/5 | 2 | 2 | 2 | 1/3 | 1 |
| Inconsistency: 0.09426 | | | | | | | | | | | | | |

**Table A9.** Comparison Matrix of 13 criteria with respect to criterion 8.

|  | $C_1$ | $C_2$ | $C_3$ | $C_6$ | $C_7$ | $C_8$ | $C_9$ | $C_{10}$ | $C_{11}$ | $C_{13}$ | $C_{15}$ | $C_{18}$ | $C_{19}$ |
|---|---|---|---|---|---|---|---|---|---|---|---|---|---|
| $C_1$ | 1 | 1/5 | 1/5 | 1/4 | 1/6 | 1/7 | 1/4 | 1/5 | 3 | 2 | 3 | 3 | 4 |
| $C_2$ | 5 | 1 | 1/4 | 1/4 | 1/2 | 1/4 | 1/3 | 1/4 | 3 | 1/2 | 4 | 1/2 | 3 |
| $C_3$ | 5 | 4 | 1 | 4 | 2 | 1/2 | 2 | 4 | 6 | 3 | 4 | 6 | 7 |
| $C_6$ | 4 | 4 | 1/4 | 1 | 4 | 1/2 | 2 | 2 | 7 | 5 | 5 | 4 | 5 |
| $C_7$ | 6 | 2 | 1/2 | 1/4 | 1 | 1/2 | 1/3 | 2 | 5 | 3 | 4 | 3 | 2 |
| $C_8$ | 7 | 4 | 2 | 2 | 2 | 1 | 3 | 2 | 8 | 5 | 8 | 6 | 7 |
| $C_9$ | 4 | 3 | 1/2 | 1/2 | 3 | 1/3 | 1 | 2 | 7 | 3 | 7 | 3 | 5 |
| $C_{10}$ | 5 | 4 | 1/4 | 1/2 | 1/2 | 1/2 | 1/2 | 1 | 5 | 3 | 4 | 3 | 5 |
| $C_{11}$ | 1/3 | 1/3 | 1/6 | 1/7 | 1/5 | 1/8 | 1/7 | 1/5 | 1 | 1/3 | 1/2 | 1/4 | 1/2 |
| $C_{13}$ | 1/2 | 2 | 1/3 | 1/5 | 1/3 | 1/5 | 1/3 | 3 | 3 | 1 | 4 | 1/2 | 1/2 |
| $C_{15}$ | 1/3 | 1/4 | 1/4 | 1/5 | 1/4 | 1/8 | 1/7 | 1/4 | 2 | 1/4 | 1 | 1/2 | 1/3 |
| $C_{18}$ | 1/3 | 2 | 1/6 | 1/4 | 1/3 | 1/6 | 1/3 | 1/3 | 4 | 2 | 2 | 1 | 2 |
| $C_{19}$ | 1/4 | 1/3 | 1/7 | 1/5 | 1/2 | 1/7 | 1/5 | 1/5 | 2 | 2 | 3 | 1/2 | 1 |
| Inconsistency: 0.09230 | | | | | | | | | | | | | |

**Table A10.** Comparison Matrix of 13 criteria with respect to criterion 9.

| | $C_1$ | $C_2$ | $C_3$ | $C_6$ | $C_7$ | $C_8$ | $C_9$ | $C_{10}$ | $C_{11}$ | $C_{13}$ | $C_{15}$ | $C_{18}$ | $C_{19}$ |
|---|---|---|---|---|---|---|---|---|---|---|---|---|---|
| $C_1$ | 1 | 3 | 3 | 1/2 | 3 | 2 | 6 | 3 | 4 | 2 | 2 | 4 | 2 |
| $C_2$ | 1/3 | 1 | 3 | 1/2 | 2 | 2 | 5 | 3 | 3 | 2 | 4 | 2 | 1/2 |
| $C_3$ | 1/3 | 1/3 | 1 | 1/3 | 1/2 | 1/2 | 2 | 1/3 | 2 | 1/3 | 1/2 | 3 | 1/4 |
| $C_6$ | 2 | 2 | 3 | 1 | 1/2 | 2 | 4 | 2 | 4 | 3 | 2 | 5 | 2 |
| $C_7$ | 1/3 | 1/2 | 2 | 2 | 1 | 1/2 | 2 | 1/3 | 2 | 1/2 | 3 | 2 | 1/3 |
| $C_8$ | 1/2 | 1/2 | 2 | 1/2 | 2 | 1 | 3 | 1/2 | 3 | 1/2 | 2 | 3 | 1/2 |
| $C_9$ | 1/6 | 1/5 | 1/2 | 1/4 | 1/2 | 1/3 | 1 | 1/4 | 1/2 | 1/3 | 1/2 | 1/2 | 1/4 |
| $C_{10}$ | 1/3 | 1/3 | 3 | 1/2 | 3 | 2 | 4 | 1 | 2 | 3 | 1/2 | 4 | 1/3 |
| $C_{11}$ | 1/4 | 1/3 | 1/2 | 1/4 | 1/2 | 1/3 | 2 | 1/2 | 1 | 1/2 | 1/3 | 1/2 | 1/5 |
| $C_{13}$ | 1/2 | 1/2 | 3 | 1/3 | 2 | 2 | 3 | 1/3 | 2 | 1 | 3 | 1/2 | 2 |
| $C_{15}$ | 1/2 | 1/4 | 2 | 1/2 | 1/3 | 1/2 | 2 | 2 | 3 | 1/3 | 1 | 3 | 1/3 |
| $C_{18}$ | 1/4 | 1/2 | 1/3 | 1/5 | 1/2 | 1/3 | 2 | 1/4 | 2 | 2 | 1/3 | 1 | 1/2 |
| $C_{19}$ | 1/2 | 2 | 4 | 2 | 3 | 2 | 4 | 3 | 5 | 1/2 | 3 | 2 | 1 |
| | | | | | | Inconsistency: 0.09890 | | | | | | | |

**Table A11.** Comparison Matrix of 13 criteria with respect to criterion 10.

| | $C_1$ | $C_2$ | $C_3$ | $C_6$ | $C_7$ | $C_8$ | $C_9$ | $C_{10}$ | $C_{11}$ | $C_{13}$ | $C_{15}$ | $C_{18}$ | $C_{19}$ |
|---|---|---|---|---|---|---|---|---|---|---|---|---|---|
| $C_1$ | 1 | 3 | 3 | 1/2 | 3 | 2 | 7 | 3 | 4 | 3 | 2 | 4 | 2 |
| $C_2$ | 1/3 | 1 | 3 | 1/2 | 2 | 2 | 5 | 3 | 3 | 2 | 4 | 2 | 1/2 |
| $C_3$ | 1/3 | 1/3 | 1 | 1/3 | 1/2 | 1/2 | 2 | 1/3 | 2 | 1/3 | 1/2 | 3 | 1/4 |
| $C_6$ | 2 | 2 | 3 | 1 | 1/2 | 2 | 4 | 2 | 4 | 3 | 2 | 4 | 2 |
| $C_7$ | 1/3 | 1/2 | 2 | 2 | 1 | 1/2 | 2 | 1/3 | 2 | 1/2 | 3 | 2 | 1/3 |
| $C_8$ | 1/2 | 1/2 | 2 | 1/2 | 2 | 1 | 3 | 1/2 | 3 | 1/2 | 2 | 3 | 1/2 |
| $C_9$ | 1/7 | 1/5 | 1/2 | 1/4 | 1/2 | 1/3 | 1 | 1/4 | 1/2 | 1/3 | 1/2 | 1/2 | 1/4 |
| $C_{10}$ | 1/3 | 1/3 | 3 | 1/2 | 3 | 2 | 4 | 1 | 2 | 3 | 1/2 | 4 | 1/3 |
| $C_{11}$ | 1/4 | 1/3 | 1/2 | 1/4 | 1/2 | 1/3 | 2 | 1/2 | 1 | 1/3 | 1/3 | 1/2 | 1/5 |
| $C_{13}$ | 1/3 | 1/2 | 3 | 1/3 | 2 | 2 | 3 | 1/3 | 3 | 1 | 3 | 1/2 | 2 |
| $C_{15}$ | 1/2 | 1/4 | 2 | 1/2 | 1/3 | 1/2 | 2 | 2 | 3 | 1/3 | 1 | 3 | 1/3 |
| $C_{18}$ | 1/4 | 1/2 | 1/3 | 1/4 | 1/2 | 1/3 | 2 | 1/4 | 2 | 2 | 1/3 | 1 | 1/4 |
| $C_{19}$ | 1/2 | 2 | 4 | 2 | 3 | 2 | 4 | 3 | 5 | 1/2 | 3 | 4 | 1 |
| | | | | | | Inconsistency: 0.09964 | | | | | | | |

**Table A12.** Comparison Matrix of 13 criteria with respect to criterion 11.

|  | $C_1$ | $C_2$ | $C_3$ | $C_6$ | $C_7$ | $C_8$ | $C_9$ | $C_{10}$ | $C_{11}$ | $C_{13}$ | $C_{15}$ | $C_{18}$ | $C_{19}$ |
|---|---|---|---|---|---|---|---|---|---|---|---|---|---|
| $C_1$ | 1 | 2 | 3 | 3 | 1/4 | 1/2 | 1/2 | 1/4 | 1/3 | 1/3 | 1/2 | 1/3 | 1/2 |
| $C_2$ | 1/2 | 1 | 1/2 | 2 | 1/4 | 1/3 | 1/2 | 1/4 | 1/2 | 1/7 | 1/4 | 1/5 | 1/3 |
| $C_3$ | 1/3 | 2 | 1 | 3 | 1/5 | 1/4 | 2 | 1/2 | 3 | 1/6 | 1/3 | 1/3 | 1/6 |
| $C_6$ | 1/3 | 1/2 | 1/3 | 1 | 1/5 | 1/8 | 1/2 | 1/5 | 1/3 | 1/8 | 1/6 | 1/4 | 1/3 |
| $C_7$ | 4 | 4 | 5 | 5 | 1 | 2 | 5 | 3 | 3 | 1/2 | 3 | 2 | 2 |
| $C_8$ | 2 | 3 | 4 | 8 | 1/2 | 1 | 5 | 2 | 1/2 | 1/3 | 2 | 2 | 3 |
| $C_9$ | 2 | 2 | 1/2 | 2 | 1/5 | 1/5 | 1 | 1/2 | 2 | 1/8 | 1/2 | 1/7 | 1/5 |
| $C_{10}$ | 4 | 4 | 2 | 5 | 1/3 | 1/2 | 2 | 1 | 1/3 | 1/6 | 1/2 | 1/2 | 2 |
| $C_{11}$ | 3 | 2 | 1/3 | 3 | 1/3 | 2 | 1/2 | 3 | 1 | 1/5 | 1/2 | 1/3 | 1/2 |
| $C_{13}$ | 3 | 7 | 6 | 8 | 2 | 3 | 8 | 6 | 5 | 1 | 4 | 2 | 4 |
| $C_{15}$ | 2 | 4 | 3 | 6 | 1/3 | 1/2 | 2 | 2 | 2 | 1/4 | 1 | 1/2 | 1/2 |
| $C_{18}$ | 3 | 5 | 3 | 4 | 1/2 | 1/2 | 7 | 5 | 3 | 1/2 | 2 | 1 | 2 |
| $C_{19}$ | 2 | 3 | 6 | 3 | 1/2 | 1/3 | 5 | 4 | 2 | 1/4 | 2 | 1/2 | 1 |
| | | | | | | Inconsistency: 0.09801 | | | | | | | |

**Table A13.** Comparison Matrix of 13 criteria with respect to criterion 13.

|  | $C_1$ | $C_2$ | $C_3$ | $C_6$ | $C_7$ | $C_8$ | $C_9$ | $C_{10}$ | $C_{11}$ | $C_{13}$ | $C_{15}$ | $C_{18}$ | $C_{19}$ |
|---|---|---|---|---|---|---|---|---|---|---|---|---|---|
| $C_1$ | 1 | 4 | 4 | 1/2 | 3 | 2 | 7 | 4 | 6 | 3 | 2 | 4 | 3 |
| $C_2$ | 1/4 | 1 | 3 | 1/2 | 2 | 2 | 4 | 3 | 3 | 2 | 4 | 2 | 1/2 |
| $C_3$ | 1/4 | 1/3 | 1 | 1/3 | 3 | 1/2 | 2 | 1/3 | 2 | 1/2 | 1/2 | 2 | 1/2 |
| $C_6$ | 2 | 2 | 3 | 1 | 3 | 2 | 4 | 2 | 4 | 3 | 2 | 5 | 2 |
| $C_7$ | 1/3 | 1/2 | 1/3 | 1/3 | 1 | 1/2 | 2 | 1/3 | 2 | 1/2 | 3 | 2 | 1/3 |
| $C_8$ | 1/2 | 1/2 | 2 | 1/2 | 2 | 1 | 3 | 1/2 | 3 | 1/2 | 2 | 3 | 1/2 |
| $C_9$ | 1/7 | 1/4 | 1/2 | 1/4 | 1/2 | 1/3 | 1 | 1/4 | 1/2 | 1/3 | 1/2 | 1/2 | 1/4 |
| $C_{10}$ | 1/4 | 1/3 | 3 | 1/2 | 3 | 2 | 4 | 1 | 2 | 3 | 1/2 | 4 | 1/3 |
| $C_{11}$ | 1/6 | 1/3 | 1/2 | 1/4 | 1/2 | 1/3 | 2 | 1/2 | 1 | 1/2 | 1/3 | 1/2 | 1/5 |
| $C_{13}$ | 1/3 | 1/2 | 2 | 1/3 | 2 | 2 | 3 | 1/3 | 2 | 1 | 3 | 1/2 | 2 |
| $C_{15}$ | 1/2 | 1/4 | 2 | 1/2 | 1/3 | 1/2 | 2 | 2 | 3 | 1/3 | 1 | 3 | 1/3 |
| $C_{18}$ | 1/4 | 1/2 | 1/2 | 1/5 | 1/2 | 1/3 | 2 | 1/4 | 2 | 2 | 1/3 | 1 | 1/2 |
| $C_{19}$ | 1/3 | 2 | 2 | 2 | 3 | 2 | 4 | 3 | 5 | 1/2 | 3 | 2 | 1 |
| | | | | | | Inconsistency: 0.09084 | | | | | | | |

**Table A14.** Comparison Matrix of 13 criteria with respect to criterion 15.

|          | $C_1$ | $C_2$ | $C_3$ | $C_6$ | $C_7$ | $C_8$ | $C_9$ | $C_{10}$ | $C_{11}$ | $C_{13}$ | $C_{15}$ | $C_{18}$ | $C_{19}$ |
|----------|-------|-------|-------|-------|-------|-------|-------|----------|----------|----------|----------|----------|----------|
| $C_1$    | 1     | 5     | 4     | 3     | 1/4   | 1/4   | 1/3   | 1/4      | 4        | 1/5      | 1/4      | 1/3      | 1/2      |
| $C_2$    | 1/5   | 1     | 1/3   | 3     | 1/5   | 1/7   | 1/3   | 1/5      | 1/4      | 1/8      | 1/4      | 1/6      | 1/4      |
| $C_3$    | 1/4   | 3     | 1     | 3     | 1/5   | 1/4   | 2     | 1/2      | 3        | 1/6      | 1/3      | 1/3      | 1/6      |
| $C_6$    | 1/3   | 1/3   | 1/3   | 1     | 1/5   | 1/8   | 1/2   | 1/5      | 1/3      | 1/8      | 1/6      | 1/8      | 1/3      |
| $C_7$    | 4     | 5     | 5     | 5     | 1     | 3     | 6     | 3        | 6        | 1/2      | 4        | 2        | 3        |
| $C_8$    | 4     | 7     | 4     | 8     | 1/3   | 1     | 5     | 2        | 5        | 1/3      | 2        | 2        | 3        |
| $C_9$    | 3     | 3     | 1/2   | 2     | 1/6   | 1/5   | 1     | 1/2      | 2        | 1/8      | 1/2      | 1/7      | 1/5      |
| $C_{10}$ | 4     | 5     | 2     | 5     | 1/3   | 1/2   | 2     | 1        | 3        | 1/6      | 1/2      | 1/2      | 2        |
| $C_{11}$ | 1/4   | 4     | 1/3   | 3     | 1/6   | 1/5   | 1/2   | 1/3      | 1        | 1/8      | 1/2      | 1/5      | 1/4      |
| $C_{13}$ | 5     | 8     | 6     | 8     | 2     | 3     | 8     | 6        | 8        | 1        | 4        | 2        | 4        |
| $C_{15}$ | 4     | 4     | 3     | 6     | 1/4   | 1/2   | 2     | 2        | 2        | 1/4      | 1        | 1/2      | 1/2      |
| $C_{18}$ | 3     | 6     | 3     | 8     | 1/2   | 1/2   | 7     | 5        | 5        | 1/2      | 2        | 1        | 2        |
| $C_{19}$ | 2     | 4     | 6     | 3     | 1/3   | 1/3   | 5     | 4        | 4        | 1/4      | 2        | 1/2      | 1        |
| Inconsistency: 0.08784 |

**Table A15.** Comparison Matrix of 13 criteria with respect to criterion 18.

|          | $C_1$ | $C_2$ | $C_3$ | $C_6$ | $C_7$ | $C_8$ | $C_9$ | $C_{10}$ | $C_{11}$ | $C_{13}$ | $C_{15}$ | $C_{18}$ | $C_{19}$ |
|----------|-------|-------|-------|-------|-------|-------|-------|----------|----------|----------|----------|----------|----------|
| $C_1$    | 1     | 2     | 3     | 3     | 1/4   | 1/5   | 1/5   | 1/4      | 4        | 1/5      | 1/4      | 1/3      | 1/2      |
| $C_2$    | 1/2   | 1     | 1/3   | 3     | 1/5   | 1/7   | 1/3   | 1/5      | 1/4      | 1/8      | 1/4      | 1/6      | 1/4      |
| $C_3$    | 1/3   | 3     | 1     | 3     | 1/5   | 1/4   | 2     | 1/2      | 3        | 1/6      | 1/3      | 1/3      | 1/6      |
| $C_6$    | 1/3   | 1/3   | 1/3   | 1     | 1/4   | 1/7   | 1/2   | 1/5      | 1/3      | 1/7      | 1/5      | 1/6      | 1/4      |
| $C_7$    | 4     | 5     | 5     | 4     | 1     | 3     | 6     | 3        | 6        | 1/2      | 4        | 2        | 3        |
| $C_8$    | 5     | 7     | 4     | 7     | 1/3   | 1     | 5     | 2        | 5        | 1/3      | 2        | 2        | 3        |
| $C_9$    | 5     | 3     | 1/2   | 2     | 1/6   | 1/5   | 1     | 1/2      | 2        | 1/8      | 1/2      | 1/7      | 1/5      |
| $C_{10}$ | 5     | 5     | 2     | 5     | 1/3   | 1/2   | 2     | 1        | 3        | 1/6      | 1/2      | 1/2      | 2        |
| $C_{11}$ | 1/4   | 4     | 1/3   | 3     | 1/6   | 1/5   | 1/2   | 1/3      | 1        | 1/8      | 1/2      | 1/5      | 1/4      |
| $C_{13}$ | 5     | 8     | 6     | 7     | 2     | 3     | 8     | 6        | 8        | 1        | 4        | 2        | 4        |
| $C_{15}$ | 4     | 4     | 3     | 5     | 1/4   | 1/2   | 2     | 2        | 2        | 1/4      | 1        | 1/2      | 1/2      |
| $C_{18}$ | 3     | 6     | 3     | 6     | 1/2   | 1/2   | 7     | 5        | 5        | 1/2      | 2        | 1        | 2        |
| $C_{19}$ | 2     | 4     | 6     | 4     | 1/3   | 1/3   | 5     | 4        | 4        | 1/4      | 2        | 1/2      | 1        |
| Inconsistency: 0.08708 |

**Table A16.** Comparison Matrix of 13 criteria with respect to criterion 19.

| | $C_1$ | $C_2$ | $C_3$ | $C_6$ | $C_7$ | $C_8$ | $C_9$ | $C_{10}$ | $C_{11}$ | $C_{13}$ | $C_{15}$ | $C_{18}$ | $C_{19}$ |
|---|---|---|---|---|---|---|---|---|---|---|---|---|---|
| $C_1$ | 1 | 4 | 4 | 2 | 3 | 2 | 5 | 4 | 6 | 3 | 2 | 4 | 3 |
| $C_2$ | 1/4 | 1 | 3 | 1/2 | 2 | 2 | 5 | 3 | 3 | 2 | 4 | 2 | 1/2 |
| $C_3$ | 1/4 | 1/3 | 1 | 1/3 | 3 | 1/2 | 2 | 1/3 | 2 | 1/3 | 1/2 | 2 | 1/2 |
| $C_6$ | 1/2 | 2 | 3 | 1 | 3 | 2 | 4 | 2 | 4 | 3 | 2 | 5 | 2 |
| $C_7$ | 1/3 | 1/2 | 1/3 | 1/3 | 1 | 1/2 | 2 | 1/3 | 2 | 1/2 | 3 | 2 | 1/3 |
| $C_8$ | 1/2 | 1/2 | 2 | 1/2 | 2 | 1 | 3 | 1/2 | 3 | 1/2 | 2 | 3 | 1/2 |
| $C_9$ | 1/5 | 1/5 | 1/2 | 1/4 | 1/2 | 1/3 | 1 | 1/4 | 1/2 | 1/3 | 1/2 | 1/2 | 1/4 |
| $C_{10}$ | 1/4 | 1/3 | 3 | 1/2 | 3 | 2 | 4 | 1 | 2 | 3 | 1/2 | 4 | 1/3 |
| $C_{11}$ | 1/6 | 1/3 | 1/2 | 1/4 | 1/2 | 1/3 | 2 | 1/2 | 1 | 1/2 | 1/3 | 1/2 | 1/5 |
| $C_{13}$ | 1/3 | 1/2 | 3 | 1/3 | 2 | 2 | 3 | 1/3 | 2 | 1 | 3 | 1/2 | 2 |
| $C_{15}$ | 1/2 | 1/4 | 2 | 1/2 | 1/3 | 1/2 | 2 | 2 | 3 | 1/3 | 1 | 3 | 1/3 |
| $C_{18}$ | 1/4 | 1/2 | 1/2 | 1/5 | 1/2 | 1/3 | 2 | 1/4 | 2 | 2 | 1/3 | 1 | 1/2 |
| $C_{19}$ | 1/3 | 2 | 2 | 2 | 3 | 2 | 4 | 3 | 5 | 1/2 | 3 | 2 | 1 |
| | | | | | | Inconsistency: 0.09114 | | | | | | | |

## Appendix G

**Table A17.** Rating of Alternative Qualitative Criteria—Linguistic Values.

| DMs | Alternatives | Qualitative Criteria | | | | | | | | | |
|---|---|---|---|---|---|---|---|---|---|---|---|
| | | $C_6$ | $C_7$ | $C_8$ | $C_9$ | $C_{10}$ | $C_{11}$ | $C_{12}$ | $C_{13}$ | $C_{14}$ | $C_{15}$ |
| $D_1$ | $A_1$ | H | EH | H | M | H | H | VP | VH | H | H |
| | $A_2$ | VH | H | H | VH | H | H | VH | H | VH | M |
| | $A_3$ | H | H | VH | VH | H | VH | VH | H | VH | H |
| | $A_4$ | M | H | M | M | H | M | M | VP | EP | M |
| $D_2$ | $A_1$ | H | EH | M | M | H | H | P | VH | M | H |
| | $A_2$ | H | VH | M | VH | H | H | VH | H | VH | H |
| | $A_3$ | M | H | H | VH | H | M | VH | H | VH | M |
| | $A_4$ | H | M | M | M | M | M | M | VP | P | M |
| $D_3$ | $A_1$ | H | EH | M | H | VH | H | P | VH | M | H |
| | $A_2$ | EH | H | H | VH | H | H | VH | H | M | M |
| | $A_3$ | VH | H | H | VH | H | P | VH | H | H | H |
| | $A_4$ | M | M | M | M | M | H | M | P | P | M |
| $D_4$ | $A_1$ | H | EH | H | H | M | M | P | VH | M | H |
| | $A_2$ | H | H | M | H | EH | P | VH | H | M | M |
| | $A_3$ | H | VH | H | VH | H | P | VH | H | VH | M |
| | $A_4$ | M | M | M | P | M | H | M | P | P | M |

### Appendix H

**Table A18.** Rating of Alternative versus Quantitative Criteria.

| Alternatives | Quantitative Criteria | | | | | | | | |
|---|---|---|---|---|---|---|---|---|---|
| | $C_1$ | | | $C_2$ | | | $C_3$ | | |
| $A_1$ | 2001 | 2500 | 3000 | 101 | 150 | 200 | 3 | 4 | 5 |
| $A_2$ | 4001 | 4500 | 5000 | 401 | 450 | 500 | 9 | 10 | 11 |
| $A_3$ | 3001 | 3500 | 4000 | 201 | 250 | 300 | 6 | 7 | 8 |
| $A_4$ | 1001 | 1500 | 2000 | 101 | 150 | 200 | 6 | 7 | 8 |

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
