# Peer review of "Ranking Startups Using DEMATEL-ANP-Based Fuzzy PROMETHEE II"

_axioms, doi:10.3390/axioms12060528_

Round 1

Reviewer 1 Report (Previous Reviewer 2)

The manuscript can be accepted in its current form 

Author Response

 Thank you very much.

Reviewer 2 Report (Previous Reviewer 3)

 I am sorry, but the authors did not provide answers to the remarks and questions from my preview review. There are a lot of descriptions of different methods, but without deep motivation why proposed Hybrid approach is better than others? The ANP method is proposed instead of AHP but also a lack of explanation. Well know facts are detailed and described but others omitted. In my opinion, this work is not a sufficient improvement from the previous paper. In my opinion, the paper also does not meet technical journal requirements.   It is unclear if in the example the authors used AHP or ANP methods.   Summing up, the paper needs more corrections.

Author Response

Please kindly check the attached file.

Reviewer 3 Report (New Reviewer)

The work is average but may be improved by the inclusion of the following

1-Quantitative information should be provided in the abstract.

2-The conclusion should be concise and to the point indicating the

application of the work.

3-The novelty of the work should be established.

4-Refs are not updated and the following ref. The addition of this may

improve the quality of this manuscript.

Author Response

Please kindly check the attached file.

Reviewer 4 Report (New Reviewer)

Author Response

Please kindly check the attached file.

This manuscript is a resubmission of an earlier submission. The following is a list of the peer review reports and author responses from that submission.

Round 1

Reviewer 1 Report

The paper proposes a Dematel-ANP-based fuzzy Promethee II model to rank startups and examine the interrelationships between factors. First of all, the paper needs editing in English, as the level of the language used is insufficient and not appropriate for scientific work. Secondly,  the authors invest a lot of time (and space in the paper) to present the already-known classical theory of fuzzy sets. Finally, the method presented is immature, lacks novelty, and needs reconsideration as in its current form has nothing interesting to present.

The paper proposes a Dematel-ANP-based fuzzy Promethee II model to rank startups and examine the interrelationships between factors. First of all, the paper needs editing in English, as the level of the language used is insufficient and not appropriate for scientific work. Secondly,  the authors invest a lot of time (and space in the paper) to present the already-known classical theory of fuzzy sets. Finally, the method presented is immature, lacks novelty, and needs reconsideration as in its current form has nothing interesting to present.

Reviewer 2 Report

Thank you for your submission to Axioms. The few major issues in the manuscript are as:

1. Overall structure and flow: The manuscript not presented a logical and coherent structure, with a clear introduction, methods, results, and conclusion. But, there are few major changes recommended in the Introduction section which fails to address or present research questions. Also, the flow of rest of paper is not well discussed in the End of Introduction section. I suggest to address these issues so readers can understand manuscript flows smoothly, and that each section leads logically into the next.

2. Clarity and concision: The manuscript is not written in a very concise manner. However, few section of study such as middle sections of Introduction part are not much clear and concise. Also literature review is not presented in a precise manner. I suggest authors to avoid using overly complex sentences or jargon. Using of these in the manuscript makes difficult for the reader to understand. The manuscript uses passive voice in writing i suggest to use active voice wherever possible.

3. Accuracy and completeness: The manuscript is not accurate and complete however, i suggest to add few latest references from Top tier journal from the related studies such as:

--(2017)

review of multi criteria decision making (MCDM) towards sustainable renewable energy development

--(2019) 

Application of MCDM methods in sustainability engineering: A literature review 2008–2018

--(2014) 

State of art surveys of overviews on MCDM/MADM methods

--(2021) 

Review on multi-criteria decision analysis in sustainable manufacturing decision making

4. Objectivity: Objectives of the study have not mentioned in the Introduction part. Add objectives in the Introduction part and use data and evidences of previous literature to support all claims.

5. Significance: Highlight the significance of the research and its implications. Please add how this study can be helpful for Startups and academia. How this study is different from the existing literature and how this can contribute to fill the existing gaps and issues.

6. Citations: Use the latest studies and sources to support all the research gaps and evidences. 

7. Grammar and punctuation: There are many grammar and punctuation mistakes.  In revision be ensure that the manuscript is free of these errors. Major mistakes in have been observed in Introduction and Conclusion sections. 

8. Formatting: The manuscript not followed Axioms authors guidelines.

9. Conclusion: I suggest to add a strong conclusion that summarizes the main findings and implications of the research, and provides a clear take-home message for the readers. In the present form the manuscript is not addressing this issue. Also, implications of the study needs to be added in before the conclusion part. 

In the end I would say manuscript needs major changes before further consideration in the Top Tier journal like Axioms. 

Authors need to to follow journal guidelines and check the manuscript with a grammar software 

Reviewer 3 Report

 Ranking fuzzy numbers based on the areas on the left and the right sides of fuzzy number

 This paper is focused on the presentation of a hybrid model DEMATEL-ANP-based fuzzy PROMETHEE II l for evaluation and rank of startups. The authors proposed also an extension of ranking fuzzy numbers to improve the flexibility of existing ranking MCDM methods. The theoretical considerations are illustrated by the numerical case.

The abstract quite adequately reflects the content of the article. The introduction clearly states the problem being investigated. The goals of the paper are quite properly defined. The authors do not precisely notice what is the novelty and contribution of the paper. The choice of references is satisfactory.

I found this work interesting. Yet, I have several remarks which could be taken into consideration to improve the quality of the paper.

Point 1. In my opinion, the novelty and original contribution should be exactly mentioned. 

Point 2. The stronger argumentation for the proposed approach could be useful, with advantages and disadvantages, and maybe a comparison with other methods. Please try to add a short discussion. Sometimes reading the text looks like a “black box” without useful motivation.

Why this hybrid method is better than others?

Point 3. Future research should be also mentioned.

Point 4. In some parts of the papers please pay attention to the notation. For instance:

Formula (1)wrong formula for a=b, or c=d.

Formula (18) In my opinion something is missing in the formula the sum is from e=1, to h not mentioned in the formula.

Formula (21) should be rather  ~ than =. This is the sum of the infinite geometric sequence.

Summing up, the paper needs major corrections.